# `pfl-research`: simulation framework for accelerating research in Private Federated Learning

**Filip Granqvist**[*]
Apple
fgranqvist@apple.com

**Congzheng Song**
Apple

**Áine Cahill**
Apple

**Rogier van Dalen**[†]

**Martin Pelikan**
Apple

**Yi Sheng Chan**
Apple

**Xiaojun Feng**
Apple

**Natarajan Krishnaswami**
Apple

**Vojta Jina**[†]

**Mona Chitnis**[†]

## Abstract

Federated learning (FL) is an emerging machine learning (ML) training paradigm where clients own their data and collaborate to train a global model, without revealing any data to the server and other participants. Researchers commonly perform experiments in a simulation environment to quickly iterate on ideas. However, existing open-source tools do not offer the efficiency required to simulate FL on large and realistic FL datasets. We introduce `pfl-research`, a fast, modular, and easy-to-use Python framework for simulating FL. It supports Tensor-Flow, PyTorch, and non-neural network models, and is tightly integrated with state-of-the-art privacy algorithms. We study the speed of open-source FL frameworks and show that `pfl-research` is 7-72× faster than alternative open-source frameworks on common cross-device setups. Such speedup will significantly boost the productivity of the FL research community and enable testing hypotheses on realistic FL datasets that were previously too resource intensive. We release a suite of benchmarks that evaluates an algorithm's overall performance on a diverse set of realistic scenarios. The code is available on GitHub at https://github.com/apple/pfl-research.

## 1 Introduction

Federated learning (FL) is an approach to collaboratively train a machine learning (ML) model between clients, with coordination by a central server [45]. Participating clients can be either user devices connected to the internet (cross-device FL) or data centers of distinct institutions, which each store data of multiple users (cross-silo FL). The common constraint is that it is infeasible to upload the data to a central server for training because of privacy, bandwidth, or other compliance concerns [1, 2]. However, even though only statistics derived from the original data are uploaded to a central server, FL algorithms alone are not formally or practically privacy preserving [12, 29, 63]. Therefore, it is essential to combine FL with techniques to preserve the privacy of users. Deployed systems often use secure aggregation and give differential privacy (DP) guarantees to achieve this [6, 13, 81, 89]. We call such systems *private federated learning* (PFL).

(P)FL is a rapidly growing field [3], and it is unrealistic for most research to be evaluated with a real-world (P)FL deployment as most researchers do not have access to a large-scale deployment.

---

[*]Corresponding author.
[†]Work done while at Apple

38th Conference on Neural Information Processing Systems (NeurIPS 2024) Track on Datasets and Benchmarks.

Even for researchers with such access, user experience constraints [68] would typically significantly limit such evaluation. Additionally, training with real edge devices is slow and typically cannot support the extensive hyperparameter tuning that may be needed [43].

Therefore, testing hypotheses in simulation is essential for continued growth in (P)FL research. FL simulation is typically much more resource intensive than training a regular (non-federated) ML model due to the extra steps involved (see Algorithm 1), thus it is of particular importance that FL simulators be efficient and scale well.

The community has identified a comprehensive list of open problems in FL [41], and effective evaluation of proposed solutions requires high quality large-scale datasets representative of these challenges. Several realistic benchmarks have been proposed to facilitate reliable evaluation of FL algorithms (see Section 2). Unfortunately, the largest and most realistic open-source benchmark datasets for FL have not been widely adopted because of the resources required to use them. As we discuss in Section 4.1, the resource demand is exacerbated by slow simulators.

We introduce `pfl-research`, a modular and easy-to-use framework for researchers to simulate FL and PFL training that is 7-72× faster than previous FL simulators. Our framework has well defined APIs that allow an FL researcher to implement their algorithms and bundle them into components that can be shared and combined with other algorithms. `pfl-research` supports both PyTorch [67] and TensorFlow [4], and it is straightforward to integrate with other ML frameworks. Additionally, scaling up with distributed simulations is seamless with Horovod [75], including multi-host, multi-GPU scenarios. Cluster workload management and scheduling systems such as slurm [40] can be used to enable running multiple simulation runs with different configurations in parallel, e.g. for hyperparameter tuning.

Our main contributions are:

**Speed** `pfl-research` is much faster than other popular FL simulators (Section 4.1) since it simulates only the computation, not the topology, of federated learning. This can significantly increase both the productivity of (P)FL researchers and the quality of publications because it makes it feasible for researchers with limited resources to evaluate algorithms on larger datasets.

**User-friendly distributed simulations** `pfl-research` makes it easy to transition from single process to distributed simulations with zero code changes. Scale-up spans multiple dimensions: number of processes, GPUs, and machines. Multiple processes can share a GPU to increase GPU utilization. Debugging, testing, and profiling are simpler compared to alternative FL frameworks.

**Privacy integration** `pfl-research` is tightly integrated with state-of-the-art privacy accountants and mechanisms, enabling a convenient workflow for experimenting with PFL and combining it with orthogonal FL features and algorithms.

**Non-gradient-descent training** `pfl-research` is also a suitable framework for researching federated learning with models that require training algorithms beyond gradient descent, such as some classical ML algorithms. We provide implementations of federated gradient boosted decision trees (GBDTs) and federated Gaussian mixture models (GMMs).

**Diverse and unified benchmarks** To get a complete picture of where an algorithm performs well, we provide setups for benchmarking algorithms in a variety of scenarios: datasets of diverse domains, IID / non-IID, no DP / Central DP. We aim to provide the same benchmarks for both PyTorch and TensorFlow so that results are comparable across frameworks.

`pfl-research` began as an inner-source project and has been used both in research and for modelling practical use cases [7–9, 33, 44, 68, 73, 88, 92].

## 2 Related Work

There are multiple open-source efforts for realizing FL. Some focus on providing infrastructure for a production environment with real edge devices [25, 57, 58, 95], some focus on simulations for research [11, 28, 30], and some provide both [34, 50, 71, 74, 87].

There are pros and cons to each framework for simulation, but there is one validly comparable metric that is key to research velocity: speed. If an FL framework runs too slowly, researchers with constrained resources will not be able to test their hypotheses on realistic benchmarks, which can

outweigh other framework benefits. For example, [77] and [55] used FLAIR in their experiments with the original TensorFlow Federated code[3] and they both mentioned that the computational resource requirements were substantial enough for them to have to reduce the scope of their experiments. Therefore, we mainly compare previous work with `pfl-research` by implementing the same cross-device benchmarks and measuring wall-clock time to finish one simulation run; see Section 4.1 for the results.

The existing FL simulation frameworks and additional works from [16, 38, 53, 56] provide benchmarks for evaluating general FL algorithms. Additionally, [83] focuses on evaluation of cross-silo FL, and [19, 59, 86] on personalized FL. A subset of each suite of benchmarks are datasets that resemble realistic FL partitions. Unfortunately, publications have largely been focused on datasets such as CIFAR10 [48], MNIST [51], Shakespeare [60], and LibSVM [17] that are not, in our view, generated by processes that adequately represent realistic FL settings A lot of progress has been made on creating large-scale, realistic datasets for PFL, including for example [15, 18, 34, 36, 50, 54, 79]. However, the large datasets realistic for FL cannot be used unless simulations become much faster. We did a survey on most recent publications at NeurIPS 2023 [66], ICLR 2024 [39], ICML 2024 [69]. There are 162 publications that provide simulation results in federated learning. 142 out of 162 publications provide results on 1-100 clients, while only 6 out of the 162 publications consider more than 1000 clients for the empirical analysis. Appendix A.1 of [11] confirmed the same trend for previous years.

Few FL frameworks integrate security and privacy features, despite FL alone not preserving the privacy of individual clients [12, 29]. Client's privacy can be violated by: (1) a server actor inspecting gradients from individual clients before aggregation; and (2) information about clients' data being extracted from the model after deployment [76]. The second issue is commonly addressed using central differential privacy (DP) [26, 65, 89]. The first issue requires, in addition to DP, secure aggregation [10, 14, 81].

Most secure aggregation schemes merely compute a sum, and therefore do not need to be simulated, but DP mechanisms need to be included in PFL simulations to accurately reflect the effects of noisy statistics. There are several popular libraries that provide implementations of DP mechanisms to apply DP noise on generic vectors [31, 91]. However, most existing FL frameworks do not provide a simple way to plug these DP mechanisms into existing FL algorithms whilst ensuring consistency of DP parameters with FL hyperparameters in experiments.

## 3  System design

`pfl-research` is built entirely in Python and bundled into one package, making installation easy on any platform. The framework is designed to be modular such that researchers are able to select a model and an algorithm, specify additional processing such as DP, and then quickly try these out for any use case.

In this section, we highlight key design elements that differentiate `pfl-research` from alternative FL frameworks. We defer the introduction to FL and DP to Appendix A, and of the detailed system design to Appendix B, with the specific classes to which these extension points map in `pfl-research` to Appendix B.1.

In `pfl-research` simulations, the algorithm drives the training procedure by: constructing queries to execute on sampled cohorts of one or multiple federated datasets for each central iteration, defining the local optimization procedure with the assistance of a local model to execute the query for a user, defining how to use aggregated statistics to update the central state, and determining when to stop training. The output of the local optimization procedure and output of the aggregation can be postprocessed by various composable features like DP mechanisms, weighting, sparsification and compression. Algorithm 1 describe this generalized PFL simulation algorithm in detail.

We highlight these key elements that contribute to the high efficiency of `pfl-research`:

1. Only one model per worker process is initialized and preserved on the GPU at all times.

2. Model parameters are updated in-place on the single model the worker process has access to. In practice, the same instantiation is used for both local training, local evaluation, central

---

[3]Available at `https://github.com/apple/ml-flair/tree/main`

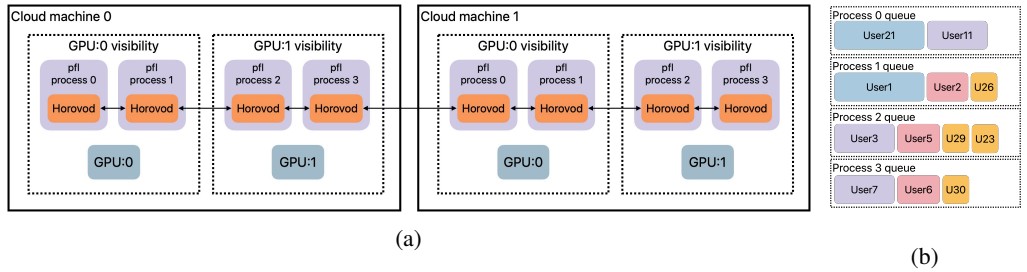

(a)

(b)

Figure 1: (a) The (simplified) architecture of distributed simulations. Each process is a replica with a different distributed context. One synchronous communication step is done to aggregate gradients and metrics. (b) Each process has a balanced queue of users to train.

   updates and central evaluation. It is only the model's state that is sometimes cloned, e.g. before training first user in new sampled cohort, but it is always cloned to already allocated tensors on the GPU. There is no memory in the order of the model size that is released and re-allocated during the course of the FL simulation.

3. `pfl-research` does not simulate the topology of FL, i.e. there is no main coordinating process to gather and broadcast model updates. Further details in Section 3.1.

4. ML framework tensors are used on the GPU end-to-end in the outer training loop. This includes DP mechanisms. Alternative frameworks, e.g. Flower, use NumPy for parts of the outer loop.

5. `pfl-research` performs load balancing of training users. This is particularly important in the case of a high dispersion in user dataset lengths, e.g. FLAIR dataset. A detailed analysis on the effect of this feature is presented in Appendix B.6.

6. Using `torch.utils.data` and `tf.data` we provide support for asynchronously loading and preprocessing entire user datasets.

`pfl-research` specifically emphasizes the flexibility for experimenting with DP. The framework is superior to existing (P)FL frameworks for simulating FL with privacy guarantees in several ways: (1) Central and local DP mechanisms and accounting methods are abstracted as pluggable components, allowing researchers to easily experiment with different combinations of DP methods; (2) Tight integration between the DP mechanisms and FL hyperparameters to prevent errors in using DP with FL, e.g. DP noise is correctly scaled according to the actual clipping bound used in each iteration; (3) DP mechanisms are implemented with GPU acceleration without data transferring between CPU and GPU; (4) `pfl-research` offers a broad range of state-of-the-art privacy mechanisms and accountants out-of-the-box for different use-cases, listed in Appendix B.5.

`pfl-research` provides interfaces that allow straightforward integration with Foundation Model (FM) frameworks such as HuggingFace [85]. The open-sourced models and datasets for FM can be easily adapted and trained in `pfl-research`, which we believe will enable and accelerate the frontier research on FM with PFL.

### 3.1 Distributed simulations

Most existing FL frameworks have explicit communication steps as they aim to simulate the topology of a real-world FL system where the central server is a bottleneck. In many deployments, the central server can be adequately scaled, and it is valuable to have a faster simulation that ignores the topology. `pfl-research` simulates only the computation of FL. Communication is necessary only if the simulation runs with multiple workers (using `Aggregator.worker_reduce`, see Appendix B.2). There is no dedicated coordinator or aggregator process needed with which all other workers communicate, unlike real-world FL. All worker processes are replicas, making debugging and transition from single-process to multi-worker training straightforward.

As depicted in Figure 1, multiple worker processes can both live on a single machine and be instantiated across multiple machines. If a machine has $g$ GPUs and $gp$ worker processes, then $p$ processes will share the same GPU. (P)FL generally has more non-GPU overhead than conventional

Table 1: Comparison of simulating CIFAR10 IID dataset with different FL frameworks. $g$ is the number of GPUs and $p$ is the number of processes concurrently training users and sharing one GPU. All experiments are run for 5 times with different random seed, and both mean and standard deviation of metrics are reported.

| Framework | $g$ | $p$ | Wall-clock time (minutes) | Accuracy | pfl-research is faster |
|---|---|---|---|---|---|
| pfl-research (PyTorch) | 1 | 1 | $10.13_{\pm 0.06}$ | $70.45\%_{\pm 0.30}$ | - |
| pfl-research (PyTorch) | 1 | 5 | $4.20_{\pm 0.10}$ | $70.25\%_{\pm 0.20}$ | - |
| pfl-research (TF) | 1 | 1 | $15.34_{\pm 0.11}$ | $70.43\%_{\pm 0.11}$ | - |
| pfl-research (TF) | 1 | 5 | $7.89_{\pm 0.26}$ | $70.52\%_{\pm 0.17}$ | - |
| FedML (PyTorch) [34] | 1 | 1 | $90.95_{\pm 0.47}$ | $71.23\%_{\pm 0.00}$ | $21.6\times$ |
| TensorFlow Federated [30] | 1 | 1 | $113.52_{\pm 1.26}$ | $70.02\%_{\pm 0.36}$ | $27\times$ |
| TensorFlow Federated [30] | 1 | 10 | $82.23_{\pm 0.49}$ | $70.12\%_{\pm 0.35}$ | $19.6\times$ |
| Flower (PyTorch) [11] | 1 | 1 | $86.88_{\pm 1.42}$ | $68.30\%_{\pm 0.12}$ | $16.4\times$ |
| Flower (PyTorch) [11] | 1 | 10 | $29.26_{\pm 0.90}$ | $67.66\%_{\pm 0.28}$ | $7\times$ |
| FedScale [50] | 1 | 1 | $425.2_{\pm 17.66}$ | $70.41\%_{\pm 0.00}$ | $101\times$ |
| FedScale [50] | 1 | 10 | $301.7_{\pm 19.22}$ | $70.90\%_{\pm 0.43}$ | $71.8\times$ |
| FLUTE [28] | 1 | 1 | $67.86_{\pm 3.36}$ | $62.26\%_{\pm 0.07}$ | $16.1\times$ |

ML training, resulting in a lower GPU utilization. Section 4.2 empirically shows that spawning more worker processes than GPUs available will increase the GPU utilization and speed up simulations.

In each central iteration, a sampled cohort of user IDs is scheduled among the worker processes. Given an associated weight for each user, `pfl-research` will do a greedy optimization based on these weights to schedule users across the worker processes to minimize stragglers. See Appendix B.6 for further details on the scheduling algorithm. Each worker process locally aggregates client model updates and metrics (while there are more clients to train for the current central iteration). Then, inter-worker aggregation is done with `all-reduce`.

## 4 Experiments

### 4.1 Comparing performance to alternative FL simulators

In this section, we present comprehensive speed benchmarking of FL simulation frameworks. We compare frameworks on two datasets: CIFAR10 IID and FLAIR. These are two of the setups also included in `pfl-research` benchmarks, described in Section 4.3. CIFAR10 simulation was done on one NVIDIA A100 GPU (40GB) and FLAIR simulation was done on four NVIDIA A100 GPUs. We refer the reader to Appendix D for details on how the frameworks were selected, the specific versions used, and additional information about the setups. For a fair comparison, the hyperparameter values are the same for different frameworks on both CIFAR10 IID and FLAIR.

In Table 1, we compare the accuracy (mainly as a consistency check) and the wall-clock time of simulation runs on CIFAR10 for the most popular open-source FL simulators. The hyper-parameters are the same for all framework setups and described in C.5. The `pfl-research` setup in PyTorch with four processes sharing the single GPU is the fastest, with a wall-clock time of only 4 minutes and 20 seconds. This means that our framework executes this task $7\times$ faster than the runner-up, Flower, and $72\times$ faster than the slowest framework, FedScale. The difference in wall-clock time is so vast because smaller scale experiments like the CIFAR10 benchmark specifically emphasizes the necessary overhead that can exist in FL simulation, and frameworks that do not minimize this overhead suffer proportionally.

As a second comparison, we use FLAIR, which instead emphasizes the efficiency of training both a larger model and dataset in distributed simulations. Due to development time constraints, we implemented the FLAIR setup for the two frameworks that are most frequently used for publications: TFF and Flower.

Table 2 shows that `pfl-research` outperforms the alternatives prominently in terms of speed on the larger scale FLAIR benchmark. Relative to `pfl-research`, TFF is more efficient on FLAIR than

Table 2: Comparison of simulating FLAIR dataset with different FL frameworks. The metric is the same as `FL-F-17 C-AP` in [79]. Note that version `0.2.0` was released after the initial open-source date of `pfl-research`, which implements improved scheduling for training users, see Appendix B.6.

| Framework | $p$ | Wall-clock time (hours) | mAP | `pfl-research` is faster |
|---|---|---|---|---|
| `pfl-research` 0.1.0 (PyTorch) | 2 | 2.16 | $61.8_{\pm 0.001}$ | - |
| `pfl-research` 0.2.0 (PyTorch) | 2 | 1.77 | $61.1_{\pm 0.3}$ | - |
| `pfl-research` 0.2.0 (PyTorch) with central DP | 2 | 1.93 | - | - |
| TensorFlow Federated [30] | 5 | 18 | $62.0_{\pm 0.3}$ | $10.2\times$ |
| Flower [11] | 5 | 28.8 | $31.5$ [4] | $16.3\times$ |

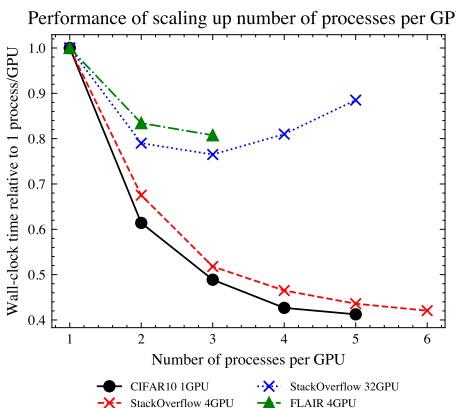

Figure 2: Speedup from scaling up number of processes per GPU in distributed simulations, while keeping the hardware resources fixed. As long as number of GPUs $\ll$ cohort size (unlike the blue line, where we use an unnecessarily large amount of GPUs given the size of model, users and cohort), the wall-clock time is monotonically decreasing when increasing number of models to train in parallel on the same GPU.

on CIFAR10, and vice versa for Flower. The third row show that enabling DP only adds $9\%$ extra simulation wall-clock time.

The number of processes per GPU, $p$, was tuned to minimize wall-clock time, and the results show that `pfl-research` needs many fewer training processes per GPU to saturate GPU utilization. Tables 1 and 2 demonstrate that before the release of `pfl-research`, the comparison of framework speed greatly depended on the choice of benchmarks.

The superior performance of `pfl-research` can be attributed to several design choices, some of which are unique to it or not commonly employed by other FL simulators, described previously in Section 3. Appendix D.4.2 show system metrics from training the CIFAR10 benchmark, giving more insight into how `pfl-research` is faster.

## 4.2 Scaling distributed training

Figure 2 shows the wall-clock time when scaling up number of processes in distributed simulations. In `pfl-research`, the number of worker processes is decoupled from the number of GPUs available, allowing processes to share a GPU. Using more than one process per GPU, $p > 1$, always has a positive effect, but the optimal number depends on the use-case (model size, cohort size). For CIFAR10 and StackOverflow [80], we can set $p = 5$ and the wall-clock time is less than $0.43$ of that with $p = 1$ with same resources, while for FLAIR we can only fit a maximum of $p = 3$ on the GPU and the relative speedup is smaller.

---

[4]Reaching parity in performance on FLAIR with Flower is still work in progress.

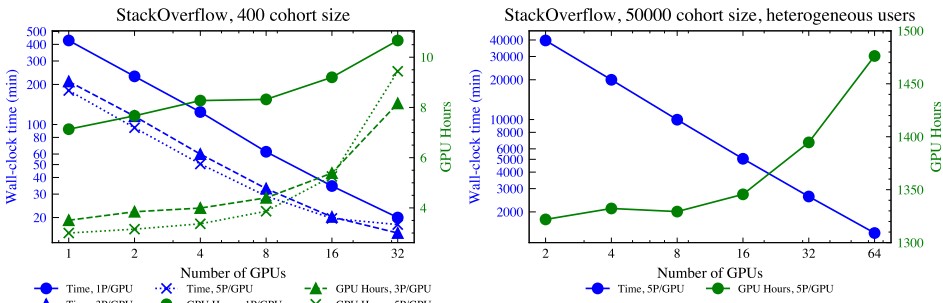

Figure 3: Speedup from scaling up distributed simulations. **Left panel:** Sweep number of processes per GPU on CIFAR10, StackOverflow and FLAIR benchmarks and keep the hardware resources pinned. **Right panel:** Sweep the number of GPUs to train the StackOverflow benchmark, repeat for 1, 3 and 5 processes per GPU. Blue lines (tracked on left y-axis) show wall-clock time. Green lines (tracked in right y-axis) show total GPU hours for the same experiments. Note that the runs with >8 GPUs use multiple hosts and thus become slightly less efficient.

Table 3: Performance of FL algorithms on the `pfl-research` benchmark suite without DP. C10 stands for accuracy on CIFAR10, SO stands for perplexity on StackOverflow, FLR stands for mean average precision (mAP) on FLAIR. The last three columns are the perplexity scores for three LLM benchmark datasets: Stanford Alpaca (SA), Aya, and OpenAssistant (OA). If not mentioned, the natural or artificial non-IID partitioning is used for the dataset, and -IID means partitioning with fixed number of samples per client drawn IID. AdaFedProx is FedProx with adaptive $\mu$ from Appendix C.3.3 in [52]. Errors for LLM experiments are detailed in Table 12
.

| Algorithm | C10-IID ↑ | C10 ↑ | SO ↓ | FLR-IID ↑ | FLR ↑ | SA ↓ | Aya ↓ | OA ↓ |
|---|---|---|---|---|---|---|---|---|
| FedAvg [60] | $70.37_{\pm.21}$ | $69.80_{\pm.35}$ | $60.87_{\pm.28}$ | $65.08_{\pm.60}$ | $61.83_{\pm.10}$ | 3.61 | 4.02 | 6.39 |
| FedProx [52] | $70.69_{\pm.33}$ | $69.79_{\pm.35}$ | $60.88_{\pm.28}$ | $64.45_{\pm1.13}$ | $62.02_{\pm.11}$ | 3.61 | 4.02 | 6.39 |
| AdaFedProx [52] | $70.39_{\pm.33}$ | $69.61_{\pm.45}$ | $61.01_{\pm.34}$ | $64.53_{\pm1.01}$ | $61.99_{\pm.11}$ | 3.61 | 4.02 | 6.40 |
| SCAFFOLD [42] | - | $60.17_{\pm1.29}$ | $62.43_{\pm.09}$ | - | $60.82_{\pm.12}$ | 3.73 | 4.77 | 6.90 |

The left panel of Figure 3 shows the wall-clock time when scaling up the number of GPUs on the StackOverflow benchmark with a cohort size of 400. The wall-clock time is reduced to 3 hours with just one GPU when scaling up number of worker processes to five, and the wall-clock time is reduced to 15 minutes when scaling up to 32 GPUs. Thus, `pfl-research` scales well with number of GPUs. The *rate* of improvement is expected to be reduced as we increase the total number of GPUs because there are fewer users per process to load balance. This is revealed by the total number of GPU hours to finish (green lines). To demonstrate a more compute intensive scenario that can be scaled up further, we increase cohort size to 50,000 and show results from scaling up in the right panel of Figure 3. We see that distributed simulations scale even better in this case, for example the increase in GPU hours when moving from 16GPUs to 32GPUs is only 3.6%.

## 4.3 Benchmarks for research

Along with the framework, we unify benchmarking of algorithms implemented in either PyTorch or TensorFlow (TF) under a collection of dataset-privacy setups, creating the matrix of benchmarks `Datasets` × {`IID, non-IID`} × {`no DP, Central DP`} × {`TF, PyTorch`} × `Algorithms`. The modularity of our framework and benchmark setups enables us to grow in any dimension: Add a dataset to benchmark all algorithms on, benchmark another algorithm, or add support for another framework or privacy mechanism. We consider the adaptive federated optimization such as FedAdam [70] as a tunable component of these algorithms, i.e. the choice of the central optimizer. The detailed hyperparameter choices for each dataset can be found in Appendix C. In this section we present the benchmark results for the PyTorch setups; we discuss the TensorFlow setups in Appendix C.1.

The initial set of algorithms we benchmark is shown in Table 3 without DP and in Table 4 with central DP. All experiments are run for five times with different random seed and metrics are averaged.

Table 4: Performance of FL algorithms on the `pfl-research` benchmark suite with central DP. The same rules and notation as in Table 3 are used. G stands for Gaussian mechanism with accounting using privacy loss distribution [22, 62] and BMF stands for banded matrix factorization mechanism [20]. Errors for LLM experiments are detailed in Table 13.

| Algorithm | DP | C10-IID ↑ | C10 ↑ | SO ↓ | FLR-IID ↑ | FLR ↑ | SA ↓ | Aya ↓ | OA ↓ |
|---|---|---|---|---|---|---|---|---|---|
| FedAvg | G | $68.77_{\pm.38}$ | $66.70_{\pm.09}$ | $69.11_{\pm.35}$ | $65.24_{\pm.22}$ | $61.49_{\pm.11}$ | 3.62 | 4.19 | 6.52 |
| FedAvg | BMF | $69.21_{\pm.51}$ | $67.07_{\pm.14}$ | $62.47_{\pm.22}$ | $65.55_{\pm.05}$ | $61.64_{\pm.10}$ | 3.62 | 4.20 | 6.53 |
| FedProx | G | $68.72_{\pm.34}$ | $66.75_{\pm.21}$ | $69.12_{\pm.34}$ | $65.24_{\pm.17}$ | $61.55_{\pm.11}$ | 3.62 | 4.20 | 6.52 |
| AdaFedProx | G | $68.78_{\pm.48}$ | $66.80_{\pm.09}$ | $69.14_{\pm.34}$ | $65.18_{\pm.16}$ | $61.58_{\pm.08}$ | 3.62 | 4.20 | 6.52 |
| SCAFFOLD | G | - | $61.31_{\pm.24}$ | $77.71_{\pm.10}$ | - | $51.64_{\pm.44}$ | 3.79 | 4.76 | 7.00 |

Evaluation is done on the validation data partitions of the original datasets without any federated splits. The common hyperparameters, such as cohort size and number of local epochs, were optimized using the FedAvg experiments. Meanwhile, specific hyperparameters for each algorithm were individually tuned. As a result, direct comparisons between the performance of different algorithms and FedAvg may not be conclusive. However, some observations based on the results include:

- Banded matrix factorization mechanism, also known as DP-FTRL when applied to FL, significantly outperforms Gaussian mechanism with PLD moments accountant on Stack-Overflow, with 10% relative improvement.

- SCAFFOLD, a popular algorithm according to citations, does not perform better than FedAvg on any of our benchmarks. [63] also report that SCAFFOLD consistently perform worse than FedAvg.

- FedProx is known for performing well on heterogeneous data partitions, but this is only marginally reflected in the FLAIR benchmark results. It is worse than FedAvg baseline on FLAIR IID, but better on FLAIR with natural heterogeneous client partitions.

The breadth of algorithms is limited at this moment, but we believe we have provided robust FedAvg baselines and a flexible foundation for future work to expand the results in terms of algorithms, complementary datasets, better hyper-parameter tuning of algorithms for more fair comparison, and other evaluation metrics, e.g. amount of communicated bits.

**LLM benchmarks** We consider three datasets for benchmarking LLM fine-tuning with `pfl-research` to demonstrate the scalability to larger models: Stanford Alpaca (SA), Aya and OpenAssistant (OA). Compared to existing FL LLM benchmarks [49, 90] where centralized datasets are artificially partitioned into users, Aya and OA have a natural user partition to simulate heterogeneity that reflects a more realistic production setting. SA contains 52,002 instruction-following data which is then artificially partitioned into users in an IID fashion. Aya contains a total of 204,112 human-annotated prompt-completion pairs, where each label has the identifier of the annotator. OA contains more than 120,000 conversational messages between human and AI assistant pairs. We collect 85,318 pairs of user inputs and assistant response with associated user identifier. We partition Aya and OA into users in a non-IID fashion using the provided user identifiers. Details of dataset partition and model fine-tuning are described in Appendix C.8.

## 5 Future work

`pfl-research` currently provides a small collection of federated algorithms. Our primary goal has been to release a fast, modular and easy-to-use simulator which is useful for advancing research in this area. We will continuously benchmark new algorithms and methods, and will rely heavily on the community for contributing implementations of new research.

We plan to carefully curate more benchmarks while maintaining a diverse set of domains for the benchmarks. All benchmark results reported in Section 4.3 used PyTorch, and releasing similar benchmarks in TensorFlow is work in progress. We recognize that all our current benchmarks

use cross-device FL. Cross-silo FL can be simulated with `pfl-research`, but we do not provide benchmarks for this scenario yet. [5]

The aggregator interface can be extended to make a client-server architecture with a dedicated aggregator and coordinator process, which might be required for testing certain research ideas not limited to synchronous FL. We also do not provide interfaces for implementing PFL with real edge devices. This is outside the scope of `pfl-research` at this time.

In some scenarios, such as large-scale cross-silo PFL simulations or training foundation models, it may be useful to train a local model for a single user (silo) on multiple GPUs. Implementation of model parallelism in pfl-research remains for future work.

Some benchmarking related to system performance left for future work are on the concept of weak scaling—to investigate the efficiency of the distributed setup as the workload and resources scale proportionally. Figure 3 had a fixed cohort size and didn't increase the cohort sizes proportionally.

## 6 Conclusion

We have introduced `pfl-research`, a fast, modular, and easy-to-use simulation framework for accelerating research in federated learning (FL) and private federated learning (PFL). It supports TensorFlow, PyTorch, and non-neural network models, and is tightly integrated with state-of-the-art privacy algorithms. It requires no code changes for scaling up distributed simulations across multiple processes, GPUs, and machines.

We compare the performance to other open-source FL simulators and our benchmarks show that `pfl-research` is 7-72× faster. The immense speedup, flexible APIs, benchmark setups with realistic datasets, integration with visualization tools, and simple debugging experience will significantly increase the productivity of (P)FL researchers. We believe that researchers will view the speedup that that `pfl-research` offers as an opportunity to validate research on more challenging and realistic datasets than what is currently common practice in the field.

### Acknowledgments and Disclosure of Funding

We thank Matt Seigel for his guidance during the early stages of this project; Kunal Talwar and Hilal Asi for valuable discussions and review of privacy features; Tatiana Likhomanenko, Anosh Raj and Kunal Talwar for valuable feedback on the paper draft; Wonhee Park, David Park, Noyan Tokgozoglu, Riyaaz Shaik, Allegra Latimer, Shams Azam, Si Beaumont, Omid Javidbakht, Tao Yu, Tatsuki Koga, Egor Shulgin, Jonny Scott and Shan Wang for giving feedback on the design and user experience of the framework; Merhawie Woldezion, Nikhil Sriraman, Tim Kolecke and Daphne Luong for helping with the process of open-sourcing `pfl-research`.

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

# Appendix

## A    Preliminaries

**Federated Learning.**    FL [60] trains an ML model with a population of users collaboratively without sharing their data with the central server. In each round of FL, the server samples a cohort of users and broadcasts the current central model to the sampled users' devices. The sampled users then train the model locally on their devices and submit their gradient updates back to the server after local training. The server aggregates the gradients received from users to produce a "pseudo-gradient" which is used to update the central model with an optimization algorithm such as SGD or Adam [70].

**Differential Privacy.**    Though users' raw data is not shared with the server in FL, the shared gradient updates can still reveal sensitive information about user data [12, 29, 63]. Differential privacy (DP) [24] can be used to provide a formal privacy guarantee to prevent such data leakage in the federated setting.

**Definition A.1** (Differential privacy).  A randomized mechanism $\mathcal{M} : \mathcal{D} \mapsto \mathcal{R}$ with a domain $\mathcal{D}$ and range $\mathcal{R}$ satisfies $(\varepsilon, \delta)$-differential privacy if for any two adjacent datasets $D, D' \in \mathcal{D}$ and for any subset of outputs $S \subseteq \mathcal{R}$ it holds that $\Pr[M(D) \in S] \leq e^{\varepsilon} \Pr[\mathcal{M}(D') \in S] + \delta$.

User-level DP guarantee is often considered in PFL [61]: $\mathcal{D}$ is the set of all possible datasets with examples associated with users, range $\mathcal{R}$ is the set of all possible models, and two datasets $D, D'$ are adjacent if $D'$ can be formed by adding or removing all of the examples associated with a single user from $D$. When a model is trained with user-level DP, one should not be able to tell whether a particular user participated in training the model or not.

A common approach to applying DP in the PFL context [61] is using Gaussian mechanism [23] where two additional steps are required in each PFL iteration: 1) the model updates from each user are clipped so that their $L_2$ norm is upper-bounded to control the sensitivity, and 2) Gaussian noise is added to the aggregated model updates from all sampled users.

Since PFL training takes multiple iterations, the privacy loss accumulates due to multiple independent accesses to the same data. To consider the total privacy loss over all iterations of training, the scale of Gaussian noise is calibrated using a privacy accountant, such as Rényi DP [64], privacy loss distribution (PLD) accountant [62] and privacy random variable (PRV) accountant [32], all of which are included in `pfl-research`. For the purpose of privacy accounting, we assume that each cohort is formed by Poisson sampling [94] where each user flips a biased coin to decide whether to participate, and that this sample is hidden from the adversary.

## B    Detailed system design

### B.1    API Design

`pfl-research` is object-oriented Python 3 code, using the following class hierarchies.

**Dataset**    `FederatedDataset` parameterizes how to partition, load, and pre-process the data of an individual user, and iterating over one produces per-user `Datasets`. The interfaces are flexible enough to support loading data from any storage using any preprocessing library. `pfl-research` also provides integration with `tf.data.Dataset` and `torch.utils.data.Dataset` (from here on referred to as "native datasets"). If the datasets of clients are small enough such that several can be loaded into memory simultaneously, a native dataset can be defined on the same level as `FederatedDataset`, returning the full data tensors of one user when requesting the next tensors. This is applicable in cross-device FL, where user datasets are typically small enough to fit into memory. If client datasets are large and loading the whole dataset at once is inefficient or requires too much memory, e.g. in large-scale cross-silo simulations, one native dataset can represent one client, with asynchronously loaded batches.

**Model**    A `Model` is an adapter for connecting a TensorFlow or PyTorch models to the simulator. Additionally, the `Model` class can be extended to implement non-neural-network models. `pfl-research` provides an implementation of federated gradient boosted decision trees (GB-DTs) [27] and federated Gaussian mixture models (GMMs).

**Algorithm**   The interface for implementing federated algorithms is `FederatedAlgorithm`. We generalize the responsibilities of an algorithm to be:

- `get_next_central_contexts` — construct configurations describing how the clients should locally train the model during the next central iteration.

- `simulate_one_user` — perform local training for one user, and return statistics and metrics.

- `process_aggregated_statistics_all_contexts` — perform an update on the central model using aggregated model update(s) and metrics.

In this way, we can implement a majority of existing FL algorithms while delegating concepts orthogonal to learning (e.g., aggregation, DP, compression) to other components that can be mixed and matched with algorithms. Algorithm 1 outlines how the above interface is used in a general FL process and Algorithm 2 describes a concrete example on how FedAvg is defined using these interfaces.

**Aggregator**   An `Aggregator` is a lightweight interface for (1) accumulating model updates from clients and (2) reducing partial aggregated model updates across worker processes. The default is accumulating with a sum, and reducing across workers with an `all-reduce` sum; see Section 3.1 for further explanation.

**Backend**   A `Backend` consumes the federated datasets, postprocessors (see below), and aggregator to implement the simulation control flow. Since `pfl-research` was released for running simulations, only the concrete backend currently included is `SimulatedBackend`.

**Postprocessor**   A `Backend` can invoke a series of `Postprocessors`. These can implement features such as compression, sparsification, and DP mechanisms by modifying local model updates as a postprocessing step, and optionally by postprocessing the aggregated model update before it is sent back to the algorithm for the central model update procedure. The server-side postprocessing steps are run in *reversed* order. It is essential to be mindful of the order of postprocessors, e.g. DP mechanisms commonly fit best as the last postprocessor for local updates so that no further modification of the sensitivity of the model update happens after clipping.

**Callback**   `TrainingProcessCallback` provides a hook into the central training loop. A callback does not have access to the data or model update and is called only after the central model has been updated, hence it should not alter the learning in any way. Some callback implementations that `pfl-research` provides are: fault-tolerant training procedure, evaluation on central dataset, exponential moving average of model, stopping criterion, reporting intermediate results (`csv` files, TensorBoard and Weights & Biases[6]), and profiling tools.

**Hyperparameters**   There are many hyperparameters to tune in (P)FL experiments, and there has been research on constructing schedules and adaptive approaches to simplify the hyperparameter tuning process [5, 47, 70]. In `pfl-research`, many hyperparameters of local model training or the algorithm can be expressed as either simple types (constant for the experiment) or as a `HyperParam` (variable across iterations). At the start of each central iteration, the algorithm requests the current value of any `HyperParam` instances to set as the static value for that iteration only. A concrete hyperparameter class can in turn extend `Postprocessor` or `TrainingProcessCallback` to hook into the training loop and adapt its value based on arbitrary rules.

**Metrics**   Metrics have conflicting implementations in various FL frameworks. We define two types below. See a concrete example for each kind in Appendix B.4.

- Central metrics — Each client returns aggregable sufficient statistics, and the metric is constructed after aggregation. This is most relevant in FL since we are interested in the performance of the central model over all data points.

- Per-user metrics — Each client produces the metric, and the aggregation over clients' results is the average of such metrics. This is more relevant when the performance of each client is of interest, e.g. in personalization and meta-learning.

**Algorithm 1** Generalized federated learning simulation.

**Input:** Initial central model $\theta_0$,
        Aggregator $agg$,
        Federated algorithm $alg$,
        Callbacks $\mathcal{B}$,
        Postprocessors $\mathcal{P}$,
        Federated datasets $\mathcal{F}$,
        A function $\mathrm{distribute\_sample}$ that samples a cohort of users from a federated dataset and distributes the cohort to workers,

1   $t \leftarrow 0$ , $stop \leftarrow False$
2   **repeat**
        `/* The algorithm generates a set of contexts C which are instructions on how to gather results from`
        `cohorts.  May modify model in preparation for this iteration.                               */`
3      $(\mathcal{C}, \theta'_t) \leftarrow alg.\mathrm{get\_next\_central\_contexts}(\theta_t, t)$
4      **if** $\mathcal{C} = \emptyset$ **then**
            `// Algorithm signaled that training should end.`
5          **break**
6      $\mathcal{S} \leftarrow \emptyset$
        `/* Sample a cohort and get aggregated results for each context.  This is done by the Backend in the`
        `implementation.                                                                              */`
7      **foreach** *central context* $c_i \in \mathcal{C}$ **do**
8          $D \leftarrow \mathrm{get\_federated\_dataset}(\mathcal{F}, c_i.\mathrm{population})$
9          **foreach** *worker* $w$ **in parallel do**
10             $\Delta_{t,i}^w \leftarrow null$
11             **foreach** *user dataset* $d_u \in \mathrm{distributed\_sample}(D, w, c_i.\mathrm{cohort\_size})$ **do**
12                $(\Delta_{t,i}^u, aux_u) \leftarrow alg.\mathrm{simulate\_one\_user}(\theta'_t, d_u, c_i)$
13                **if** $\Delta_{t,i}^u \neq null$ **then**
                  `// The results from local optimization is passed through all postprocessors, which may`
                  `manipulate the statistics and use auxiliary non-private information from the user.`
14                   **foreach** *postprocessor* $p \in \mathcal{P}$ **do**
15                     $\Delta_{t,i}^u \leftarrow p.\mathrm{postprocess\_one\_user}(\Delta_{t,i}^u, aux_u)$
16                   $\Delta_{t,i}^w \leftarrow agg.\mathrm{accumulate}(\Delta_{t,i}^w, \Delta_{t,i}^u)$
            `// Synchronize aggregates from all workers.`
17             $\Delta_{t,i} \leftarrow agg.\mathrm{worker\_reduce}(\Delta_{t,i}^w, c_i, w)$
18          **foreach** *postprocessor* $p \in \mathrm{reversed}(\mathcal{P})$ **do**
19             $\Delta_{t,i} \leftarrow p.\mathrm{postprocess\_server}(\Delta_{t,i}, c_i)$
20          $\mathcal{S} \leftarrow \mathcal{S} \cup \{\Delta_{t,i}\}$
        `/* The algorithm consumes the set of aggregated statistics from all queried cohorts and produces a new`
        `central model.                                                                               */`
21      $\theta_{t+1} \leftarrow alg.\mathrm{process\_aggregated\_statistics\_all\_contexts}(\mathcal{C}, \mathcal{S}, \theta'_t)$
22      **foreach** *callback* $b \in \mathcal{B}$ **do**
            `// A callback may force stop training, e.g.  early stopping.`
23          $stop$ |= $b.\mathrm{after\_central\_iteration}(\theta_{t+1}, t)$
24      $t \leftarrow t + 1$
25   **until** $stop$

## B.2    FL simulation

An algorithm runs for a number of central iterations, which is either determined by the algorithm or interrupted by a callback. A context $c_i \in \mathcal{C}$ is a recipe for how to run the next central iteration. It contains parameters for local model training and evaluation, the state of the algorithm, dataset target criteria, and parameters for other steps of the algorithm. A federated algorithm $alg$ operates on a group of federated datasets $\mathcal{F}$ and each context $c_i$ targets one federated dataset $D \in \mathcal{F}$.

Metrics get accumulated on almost every step in Algorithm 1 and reported at the end of each central iteration, but the processing of metrics is abstracted away for simplicity.

---

[6] `https://wandb.ai/`

Aggregation is decoupled in `pfl-research` from algorithms to provide a uniform interface for different methods of aggregating training statistics. This is separated in two sub-operations, `accumulate` and `worker_reduce`. It is essential to make the aggregator agnostic to the number of worker processes spawned, which requires a concrete aggregator implementation to adhere to the following rules:

1. A concrete aggregator derived from the abstract base class `Aggregator` must implement two functions: $f$ for `accumulate` and $g$ for `worker_reduce` where
   - $f$ represents the operation associated with accumulation of statistics from users within a worker process $w$. Upon receiving a user statistics $\Delta_u$, the worker accumulated state $S_w$ is updated as: $S_w = f(S_w, \Delta_u)$,
   - $g$ represents the operation used for inter-worker aggregation on the set of accumulated states from all workers. Each worker process $w$ calls `worker_reduce` and receives the aggregated results as $S = g(\{S_w \mid \forall \text{ worker process } w\})$.

2. For any accumulated state $S_a, S_b$ and user statistics $\Delta$ of the context over which `Aggregator` operates, the functions $f$ and $g$ must satisfy the following property:

$$\forall S_a, S_b, \Delta \quad g(\{f(S_a, \Delta), S_b\}) = g(\{f(S_b, \Delta), S_a\}) = f(g(\{S_a, S_b\}), \Delta).$$

The most common implementation in FL for $f$ and $g$ is vector summation where the state $S$ is a vector, $f(S, \Delta) = S + \Delta$ and $g(\{S_i \mid i \in [n]\}) = \sum_{i=1}^{n} S_i$. The interface provided by `pfl-research` allows customized aggregation of user statistics by adjusting $f$ and $g$ accordingly. For example, to gather the individual statistics from all users, $f$ and $g$ can be implemented as vector set union where the state $S$ is a set of vectors, $f(S, \Delta) = S \cup \{\Delta\}$ and $g(\{S_i \mid i \in [n]\}) = \cup_{i=1}^{n} S_i$.

A concrete example of the algorithm FEDAVG is given in Appendix-B.3, and of metrics in Appendix-B.4

The statistics may be modified using postprocessors before or after aggregation. The differential privacy features in `pfl-research` are implemented using postprocessors.

## B.3 Federated Averaging example

As a concrete example, pseudo-code for federated averaging (FEDAVG) is shown in Algorithm 2. Here, $\mathcal{F}$ should be defined as two populations: one group of users only for training, and the other only for validating performance. In each iteration, FEDAVG generates $\mathcal{C}$ containing two central contexts corresponding to the two populations: one context with hyper-parameters for instructing how $alg$ runs on the training population; and the other for describing how $alg$ evaluates on the validation population (e.g. not returning any model updates). In other algorithms there can be additional contexts if for example $alg$ further divides the train population into multiple clusters of users that should train different models or with different hyper-parameters. FEDAVG expects a single aggregated model update, which is averaged and applied on the central model with the central optimizer $Opt_c$.

## B.4 Metrics example

A metric can be defined on two levels in `pfl-research`: per-user and central. If user $U_1$ has 1 datapoint and user $U_2$ has 7 datapoints, where $U_1$ predicts the output correctly on its datum and $U_2$ fails to predict correct output for all data, then the accuracies are:

$$per\text{-}user\, Acc_{\{U_1, U_2\}} = \frac{1/1 + (0/7)}{2} = 0.5$$

$$central\, Acc_{\{U_1, U_2\}} = \frac{1 + 0}{8} = 0.125$$

The per-user metric has built-in weighting of how many users do well for the metric, hence per-user metrics can be useful to measure after local training in the case of personalisation. On the other hand, if the researcher is mainly interested in the performance of the central model over the entire dataset, then the central metric is the appropriate choice because it reveals that the model did not perform well on most datapoints in the example above.

---

**Algorithm 2** FEDAVG using `pfl-research` interfaces

---

**Class** *FedAvg*:

    **Input:** Number of central iterations $T$,
           Evaluation frequency $\tau$,
           Local optimizer $Opt_l$,
           Central optimizer $Opt_c$,
           Other hyper-parameters for algorithm and model $kwargs$,

1   **Function** `get_next_central_contexts`($\theta_t$, $t$):
2     **if** $t = T$ **then**
      // Training is complete
3       **return** $\emptyset, \theta_t$
4     $\mathcal{C} \leftarrow \{$create_context_for_training$(kwargs)\}$
5     **if** $t \mod \tau = 0$ **then**
6       $\mathcal{C} \leftarrow \mathcal{C} \cup \{$create_context_for_evaluation$(kwargs)\}$
    /* The model is not modified in preparation for the next central iteration.       */
7     **return** $\mathcal{C}, \theta_t$
8   **Function** `simulate_one_user`($\theta$, $d_u$, $c$):
9     **if** $c$.population $=$ Train **then**
10       $(\theta', aux) \leftarrow$ local_optimize$(\theta, d_u, Opt_l, c$.model_train_params$)$
11       $\Delta \leftarrow \theta - \theta'$
12       **return** $(\Delta, aux)$
13     **else**
14       evaluate$(\theta'_t, d, c$.model_eval_params$)$
15       **return** $(null, null)$
16   **Function** `process_aggregated_statistics_all_contexts`($\mathcal{C}$, $\mathcal{S}$, $\theta_t$):
    // Expecting $\mathcal{S}$ to be length 1, the aggregated model update, extract it.
17     $\{\Delta\} \leftarrow \mathcal{S}$ // $\Delta$ is weighted, average it.
18     $\Delta \leftarrow$ average$(\Delta)$
19     $\theta_{t+1} \leftarrow Opt_c(\theta_t, \Delta)$
20     **return** $\theta_{t+1}$

---

### B.5 Privacy features

`pfl-research` implements or integrates with state-of-the-art differential privacy methods for FL, which are available for PyTorch, Tensorflow, and non-neural network models:

- Gaussian mechanism [24].
- Gaussian mechanism with adaptive clipping [5].
- Laplace mechanism [24].
- Banded matrix factorization mechanism [20].
- Rényi DP privacy accountant [64].
- Privacy loss distribution (PLD) privacy accountant [22, 62].
- Privacy random variable (PRV) privacy accountant [32].

`pfl-research` supports simulations with both local as well as central DP guarantees. Running a DP mechanism locally results in slower simulations because the generation and addition of noise happens once per sampled user as opposed to once per central iteration. To speed up simulations, `pfl-research` implements `GaussianApproximatedPrivacyMechanism`[7], which utilizes the central limit theorem [72] to approximate a local DP mechanism as a Gaussian mechanism where the noise is applied centrally. This has the same effect as using the equivalent local mechanism, but the noise is only applied once per central iteration. This approach can be used only for simulation, and in a real-world PFL system, the local DP mechanism should be applied locally to ensure local DP guarantees.

---

[7] `https://apple.github.io/pfl-research/reference/privacy.html#module-pfl.privacy.approximate_mechanism`

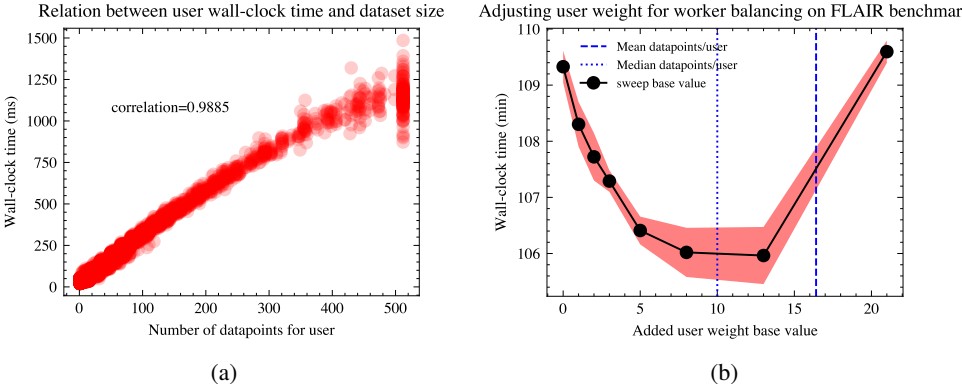

Figure 4: (a) Scatter plot of dataset size for users and the wall-clock time for training that user. Strong correlation means we can use the dataset size for scheduling the work among the worker processes. (b) Change in wall-clock time for 5000 central iterations when a fixed base value is added to each user's weight for worker scheduling.

Table 5: The maximum straggler time (difference in wall-clock time between the worker processes that finish first and last), averaged over all central iterations.

| Setup | maximum straggler time, averaged over iterations (ms) |
|---|---|
| No scheduling (uniform user split) | 1294 |
| Greedy scheduling | 484 |
| Greedy scheduling +median | 178 |

## B.6 Worker scheduling

Training users is scheduled among the worker processes. To minimize latency, the worker processes cannot pull the next user ID to train from a central queue. Therefore, the scheduling of users is pre-calculated for each cohort by iterating though users in order sorted by weight and greedily assigning the next user to the worker process with smallest total weight accumulated. The weight should represent the relative wall-clock time for training the user. We choose the number of datapoints of a user as a proxy because it has high correlation with the true wall-clock time in `pfl-research`, see Figure 4a. There is always a small overhead in FL, and if we represent this as a small base value added to all users' weights the scheduling becomes more optimal and further speeds up training. Figure 4b show that adding a base value close to the median number of datapoints per user results in an additional 3% speedup, and a total of 19% speedup compared to no worker scheduling for the FLAIR dataset (131 minutes wall-clock time).

The central iteration will only finish once all worker processes have completed their assigned training. When the first process finish training, it becomes idle until the last process is finished (straggler process). Table 5 show the maximum straggler time for our scheduling methods. Performing the greedy optimization of weights with a base value show a clear improvement in minimizing the stragglers.

To showcase a few samples, Figure 5 show the histogram of weights assigned to each worker process for a specific central iteration. The difference between the minimum and maximum value of the per-worker wall-clock time is the maximum straggler time. The disparity is clearly the smallest for greedy scheduling with a base value.

## C  Details for benchmark setups

### C.1  TensorFlow benchmarks

Compiling benchmark results for TensorFlow setups is left for future work. Currently, the CIFAR10 benchmarks and StackOverflow benchmarks have equivalent setups

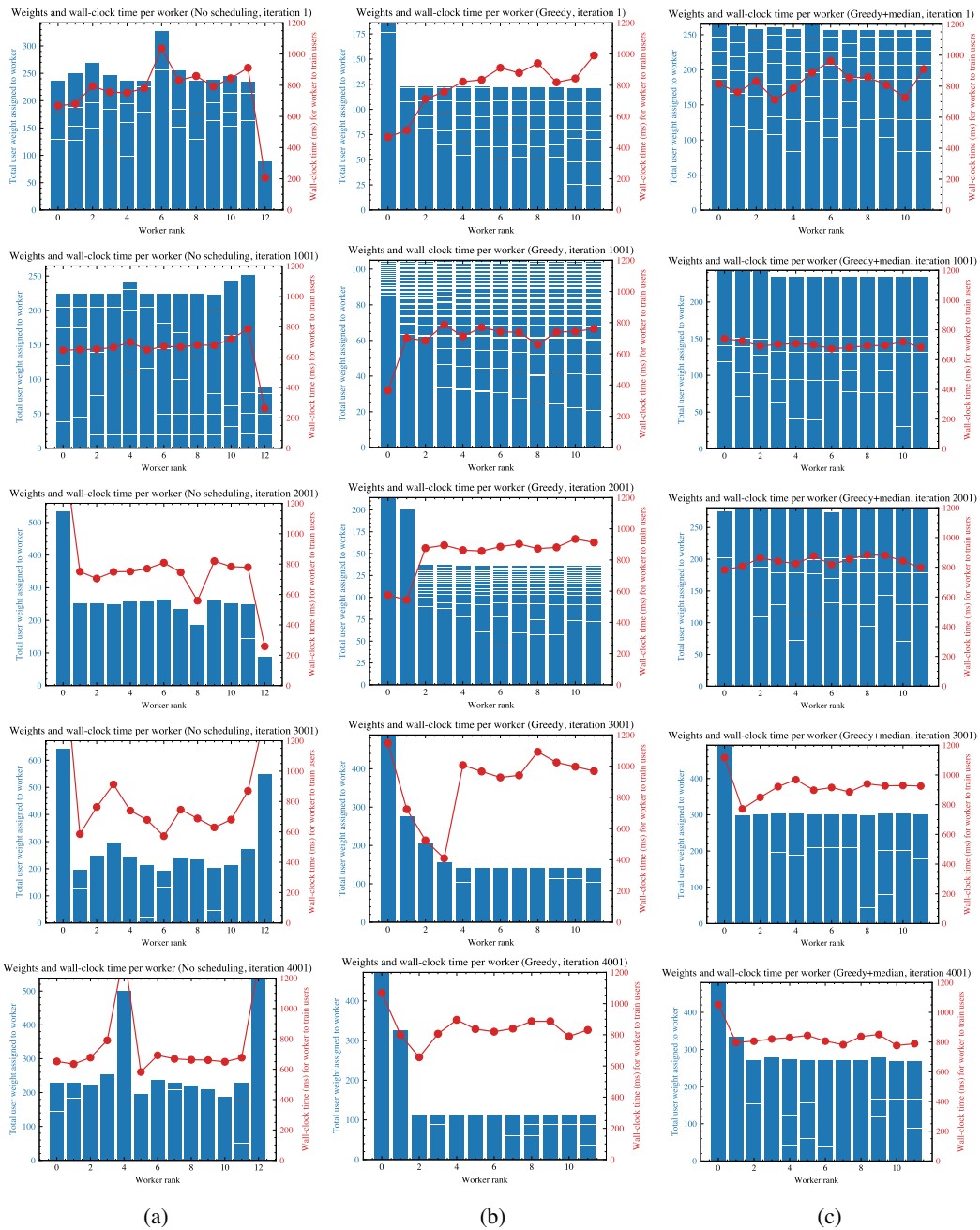

Figure 5: Histograms of user weights allocated to each worker process. The red lines show the total wall-clock times per worker process. A more evenly distributed red line means work was balanced more efficiently among worker processes. Each row show samples from a different central iteration, column a) show uniformly scheduling users, b) show scheduling with greedy optimization, c) show scheduling with greedy optimization and adding the median number of datapoints per user as base value for each weight.

Table 6: Datasets used in the original publications of the algorithms. XMNIST stands for MNIST and any of its extensions (EMNIST [21], FEMNIST [15]). Red color indicates the data generation process were not of a federated nature.

| Algorithm | Synthetic [15] | XMNIST | CIFAR10 [48] | Shakespeare [60] | Reddit [60] | Sent140 [15] |
|---|---|---|---|---|---|---|
| FedAvg [60] | | ✓ | ✓ | ✓ | ✓ | |
| FedProx [52] | ✓ | ✓ | | ✓ | | ✓ |
| AdaFedProx [52] | ✓ | | | | | |
| SCAFFOLD [42] | | ✓ | | | | |

in TensorFlow, available at `https://github.com/apple/pfl-research/tree/develop/benchmarks/image_classification/tf` and `https://github.com/apple/pfl-research/tree/develop/benchmarks/lm/tf` respectively. The FLAIR benchmark has a TensorFlow setup in `https://github.com/apple/ml-flair/tree/main/benchmark` but it is implemented in TensorFlow Federated.

## C.2 Original benchmarks of algorithms

Table 6 present the datasets that each algorithm from Table 3 benchmarked on in their original papers. It is evident that the algorithms were not sufficiently evaluated on a diverse set of realistic FL datasets.

## C.3 Secure aggregation

`pfl-research` focuses on simulating the effects of PFL. Since secure aggregation techniques merely compute a sum [10, 14], secure aggregation does not impact the outcomes of a simulation and can thus be omitted. Also, `pfl-research` does not emulate constraints and effects of a real-world deployed PFL system. Therefore, we do not include any implementations of common secure aggregation methods in the initial release of `pfl-research`.

## C.4 Central DP

Each benchmark has one version without DP and one with central DP. The current state-of-the-art PFL deployments [6, 13, 81, 89] combine central DP with a cryptographically secure aggregation to protect any individual user's model update. Local DP is therefore not generally required. Though local privacy is supported in the `pfl-research` framework, the curated PFL benchmark suite only uses central DP. The privacy parameters we use for all benchmarks are shown in Table 7.

Table 7: Parameters used for a privacy accountant in all benchmarks that use central DP.

| Parameter | value |
|---|---|
| $M$ (population) | $10^6$ |
| $\varepsilon$ | 2.0 |
| $\delta$ | $1/M = 10^{-6}$ |

Each benchmark that includes a privacy accountant has a sampling rate set to $\frac{\tilde{C}}{M}$ where $\tilde{C}$ is the cohort size and $M$ is the population size. When training in a production system on real devices, $\tilde{C}$ is typically in the order of thousands to reduce the impact of DP noise on convergence [88, 89]. To simulate the real world cross-device PFL setting with millions of devices, we set $M = 10^6$ instead of the actual number of clients in each dataset. This assumption makes sense for a simulation as long as we can expect to be able to have at least $10^6$ devices in the production use case. We also want to run the experiments faster with a smaller cohort size $C$ (in the order of hundreds) and verify the impact of the DP noise for a larger $\tilde{C}$. We refer $\tilde{C}$ as *noise cohort size* hereinafter. We introduce a variable $r = \frac{C}{\tilde{C}}$ and scale the noise standard deviation by $r$, such that the noise added to the averaged aggregates of $C$ clients will have the same scale as if the aggregates had $\tilde{C}$ clients [61]. Figure 6

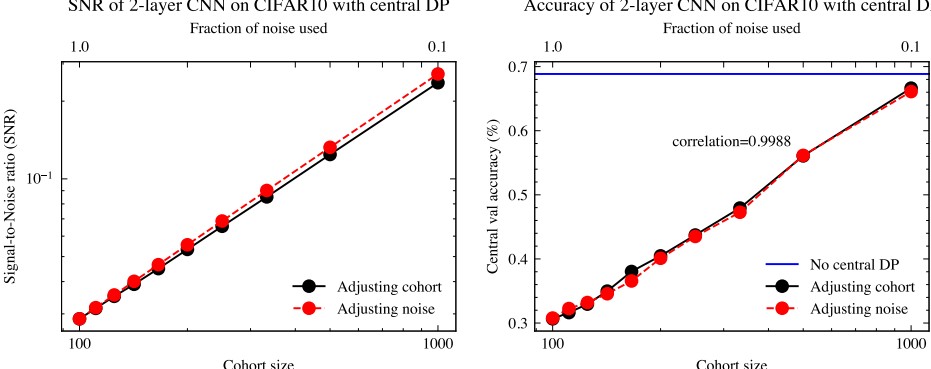

Figure 6: Performance of a 2-layer CNN on CIFAR10 for various cohort sizes $C$ (bottom x-axis, black lines) or noise scale $r$ (top x-axis, red lines). **Left panel:** signal-to-noise ratio, as defined in Equation 1. **Right panel:** same experiments, but showing accuracy on centrally held validation set.

shows the accuracy and signal-to-noise ratio (SNR) on the CIFAR10 benchmark if we either increase cohort size $\tilde{C}$ or reduce the noise with $r$. Signal-to-noise ratio is defined as

$$\text{SNR} = \frac{\|\Delta\|_2}{\sqrt{d\sigma^2}}, \tag{1}$$

where $\Delta$ is the unnoised aggregated model update, $d$ is the dimension of $\Delta$ and $\sigma$ is the standard deviation of noise from DP mechanism. The correlation of the metrics is close to $1$. This means that instead of achieving good performance on CIFAR10 with $\tilde{C} = 1000$, we can instead run faster simulations with $C = 100$ and $r = 0.1$ and expect similar results. When rescaling the noise by $r$, one must make sure for every case that:

1. $C$ is still high enough to receive good signal without noise in the simulation.

2. $\tilde{C}$ is the cohort size needed in a real PFL deployment to achieve this performance, and it should thus be a realistic value for the particular use case.

For Banded Matrix Factorization mechanism we set the minimum number of central iterations between two occurrences of the same user to $48$ for all benchmarks. This means 1 participation per day if the time for one central iteration in a real PFL system is 30 minutes. This value is not derived from any deployed PFL systems, but a conservative guess for large cross-device PFL systems.

## C.5 CIFAR10

We define four variations of CIFAR10 benchmarks: $\{\texttt{IID}, \texttt{non-IID}\} \times \{\texttt{no DP}, \texttt{central DP}\}$, such that there exist smaller and easier benchmarks with different end goals available for prototyping. The task is single-class classification of images using the CNN architecture originally defined in Table 4 of [70]. The hyper-parameters are outlined in Table 8.

Table 8: Hyper-parameters used in the four CIFAR10 benchmarks.

| Setup | Parameter | value |
|---|---|---|
| all | Central iterations | 1500 |
| | Central learning rate | 1.0 |
| | Central optimizer | SGD |
| | Cohort size $C$ | 50 |
| | Local epochs | 1 |
| | Local learning rate | 0.1 |
| | Local batch size | 10 |
| | Datapoints per user | 50 |
| | Evaluation frequency | 10 |
| | Evaluation batch size | 10000 |
| non-IID | Dirichlet alpha | 0.1 |
| central DP | clipping bound | 0.4 |
| | Noise cohort size $\tilde{C}$ | 1000 |

The data is split into $50000/50 = 1000$ users to resemble a cross-device population. As explained in Appendix C.4 we simulate with cohort size $C$, but manually reduce the noise by a fraction $\frac{C}{\tilde{C}}$ to simulate the effect of running with a larger cohort size. `pfl-research` supports setting a different batch size for training and evaluation. It is a good practice to set the evaluation batch size to the largest value that the GPUs can handle, which will speed up simulations.

The code can be found in `benchmarks/image_classification`[8].

## C.6 StackOverflow

The StackOverflow dataset is licensed under the Creative Commons Attribution-ShareAlike 3.0 Unported License. We define two variations of benchmarks on StackOverflow [80]: {no DP, central DP}. No dataset partitioning is needed as the dataset has inherent user identifiers. The task is next-word prediction using a transformer model with 1,962,912 parameters. The hyper-parameters are outlined in Table 9.

---

[8]`https://github.com/apple/pfl-research/tree/develop/benchmarks/image_classification`. Run main script with `--help` for further description on hyper-parameters.

Table 9: Hyper-parameters used in the two StackOverflow benchmarks.

| Setup | Parameter | value |
|---|---|---|
| all | Central iterations | 2000 |
| | Central learning rate | 0.1 |
| | Central lr warmup | 50 |
| | Central optimizer | Adam |
| | Adam adaptivity degree | 0.1 [70] |
| | Adam $beta_1$ | 0.9 [70] |
| | Adam $beta_2$ | 0.99 [70] |
| | Cohort size $C$ | 400 |
| | Local epochs | 1 |
| | Local learning rate | 0.3 |
| | Local batch size | 16 |
| | Evaluation frequency | 20 |
| | Evaluation batch size | 1024 |
| | Central eval data size | 1% |
| | Max sentences per user | 64 [84] |
| | Sequence length | 20 [70, 84] |
| | Max tokens per user | 1600 [61] |
| | Minimum datapoints per user | 1 |
| | Model embedding size | 96 |
| | Model num heads | 8 |
| | Model encoder feedforward size | 1536 |
| | Model transformer layers | 3 |
| | Dropout rate | 0.1 |
| central DP | clipping bound | 1.0 |
| | Noise cohort size $\tilde{C}$ | 5000 |

In this benchmark (and all following benchmarks) Adam optimizer for central updates is used, with modified hyper-parameters suggested by [70] for stable learning with FL. The performance of the model increase slightly if trained beyond 2000 iterations, but we fix the iterations such that converging with a fewer amount of iterations is also rewarded in the benchmark.

The code can be found in `benchmarks/lm`[9].

## C.7 FLAIR

The annotations of FLAIR are licensed under CC-BY-NC 4.0, while each image has a permissive license specified by each individual author[10]. We define four variations of FLAIR benchmarks: $\{IID, non\text{-}IID\} \times \{no\ DP, central\ DP\}$. The inherent user partitions in FLAIR display strong heterogeneity, as discussed in [79]. It can therefore be particularly insightful to compare performances between original partitioning and IID partitioning. The original FLAIR benchmarks include setups for multi-class image classification of coarse-grained labels, fine-grained labels, pre-trained ResNet18 [35] and train from scratch. For simplicity, our benchmark suite only include the coarse-grained labels with pre-trained ResNet18, but we encourage researchers to also consider the other setups when a wider evaluation of image classification tasks is of particular interest. The hyper-parameters are outlined in Table 10.

---

[9] `https://github.com/apple/pfl-research/tree/develop/benchmarks/lm`. Run main script with `--help` for further description on hyper-parameters.

[10] `https://github.com/apple/ml-flair/blob/main/ATTRIBUTION.txt`

Table 10: Hyper-parameters used in the FLAIR benchmarks.

| Setup | Parameter | value |
|---|---|---|
| all | Central iterations | 5000 |
| | Central learning rate | 0.1 |
| | Central optimizer | Adam |
| | Adam adaptivity degree | 0.1 [70] |
| | Adam $beta_1$ | 0.9 [70] |
| | Adam $beta_2$ | 0.99 [70] |
| | Cohort size $C$ | 200 |
| | Local epochs | 2 |
| | Local learning rate | 0.01 |
| | Local batch size | 16 |
| | Evaluation frequency | 20 |
| | Evaluation batch size | 512 |
| | Pre-trained | true |
| | Max images per user | 512 |
| IID | Datapoints per user | 50 |
| central DP | clipping bound | 0.1 |
| | Noise cohort size $\tilde{C}$ | 5000 |

The code can be found in benchmarks/flair[11].

## C.8 LLM

Table 11: Hyper-parameters used in the LLM benchmarks. SA denotes Stanford Alpaca and OA denotes OpenAssistant.

| Setup | Parameter | SA | Aya | OA |
|---|---|---|---|---|
| all | Central iterations | | 1000 | |
| | Central learning rate | | 0.01 | |
| | Central optimizer | | Adam | |
| | Adam adaptivity degree | | $10^{-4}$ | |
| | Adam $beta_1$ | | 0.9 | |
| | Adam $beta_2$ | | 0.99 | |
| | Cohort size $C$ | | 100 | |
| | Local epochs | | 1 | |
| | Local learning rate | 0.01 | | 0.1 |
| | Local batch size | | 4 | |
| | Evaluation frequency | | 10 | |
| | Evaluation batch size | | 12 | |
| | Max texts per user | - | | 64 |
| central DP | clipping bound | | 0.1 | |
| | Noise cohort size $\tilde{C}$ | | 5000 | |

We consider three datasets for LLM benchmarks: Stanford Alpaca [82], Aya [78] and OpenAssistant [46]. All three datasets are available on HuggingFace Dataset Hub [12]. We define two variations of LLM benchmarks: {no DP, central DP}. Aya and OpenAssistant datasets have inherent user partition which is ideal for simulating fine-tuning LLMs in the federated setting. For Stanford Alpaca which does not have user partition, we first sample the length $L$ of each user dataset using Poisson distribution with expectation of 16 data per user, and then assign the unused $L$ data points to the user. The partition stops when all data points are assigned. For Aya, we constrain the maximum

---
[11]https://github.com/apple/pfl-research/tree/develop/benchmarks/flair. Run main script with -help for further description on hyper-parameters.
[12]https://huggingface.co/datasets.

Table 12: Performance of FL algorithms on LLM benchmarks without DP.

| Algorithm | Alpaca | Aya | OASST |
|---|---|---|---|
| FedAvg | $3.61\pm_{.0022}$ | $4.02\pm_{.0103}$ | $6.39\pm_{.0151}$ |
| FedProx | $3.61\pm_{.0021}$ | $4.02\pm_{.0053}$ | $6.39\pm_{.0190}$ |
| AdaFedProx | $3.61\pm_{.0022}$ | $4.02\pm_{.0061}$ | $6.40\pm_{.0191}$ |
| SCAFFOLD | $3.73\pm_{.0020}$ | $4.77\pm_{.0157}$ | $6.90\pm_{.0240}$ |

Table 13: Performance of FL algorithms on LLM benchmarks with DP.

| Algorithm | DP | Alpaca | Aya | OASST |
|---|---|---|---|---|
| FedAvg | G | $3.62\pm_{.0025}$ | $4.19\pm_{.0156}$ | $6.52\pm_{.0216}$ |
| FedAvg | BMF | $3.62\pm_{.0023}$ | $4.20\pm_{.0165}$ | $6.53\pm_{.0052}$ |
| FedProx | G | $3.62\pm_{.0026}$ | $4.20\pm_{.0137}$ | $6.52\pm_{.0231}$ |
| AdaFedProx | G | $3.62\pm_{.0026}$ | $4.20\pm_{.0151}$ | $6.52\pm_{.0244}$ |
| SCAFFOLD | G | $3.79\pm_{.0035}$ | $4.76\pm_{.0181}$ | $7.00\pm_{.0100}$ |

data per user to 64, and if an annotator has more than 64 pairs, we evenly split this annotator's data into smaller subsets of size 64. As edge devices have limited memory and compute, we choose the TinyLlama-1.1B [93] for benchmarking and fine-tune it on all three datasets with LoRA [37] rank of 8 and mixed-precision training using bfloat16. We report the final perplexity as the utility metric. The hyper-parameters are outlined in Table 11.

# D  Benchmarking alternative FL frameworks

We selected FL simulators for comparison that are popular (at least 400 stars on GitHub), actively maintained (at least 1 commit in the main branch in the past 3 months), and primarily meant for simulations (as opposed to live deployments). We also included FLUTE and FedScale since they prioritize speed and scalability.

Even though it is not a realistic federated dataset, we chose CIFAR10 for the initial round of comparing framework speed because it is widely popular in FL publications and the frameworks had existing CIFAR10 setups implemented with best practices in the respective repository. We adapted each initial CIFAR10 setup to match the IID setup in C.5. This is an ideal setup for benchmarking speed because the model is small; the IID dataset split is simple; central evaluation is lightweight; and training only utilizes 1 GPU with minimal setup, e.g. 5 local iterations and no DP.

We only benchmarked `pfl-research`, TensorFlow Federated and Flower on FLAIR because of time constraints. Implementing the metrics used in FLAIR for all different frameworks was particularly time consuming.

To ensure no framework included any improvements that could be influenced by `pfl-research`, we used the last commit before the release date of `pfl-research`, 29th February, for each framework. We benchmarked `pfl-research` with version 0.1 and did not include no additional speedup improvements merged after 1st February 2024.

## D.1  FedML

We used the commit `28cd33b2d64fda1533c1bac6109c14f13c1012f7` from 29th February 2024 in our CIFAR10 FedML setup and adopted the existing CIFAR10 setup[13]. We changed the model and hyper-parameters to reflect Appendix C.5. Additionally, we did the following modifications in FedML to more accurately reflect Appendix C.5:

- Remove random crop in CIFAR10 data loader.
- Remove random horizontal flip in CIFAR10 data loader.

---

[13]`https://github.com/FedML-AI/FedML/tree/28cd33b2d64fda1533c1bac6109c14f13c1012f7/python/examples/federate/simulation/sp_fedavg_cifar10_resnet56_example`

- Only evaluate 1 user as central evaluation as each user has a copy of all the test data.

## D.2 Flower

We used the commit `f25ddc13727ee85f372583f6d85c046803a76329` from 29th February 2024 in our CIFAR10 and FLAIR setup with Flower and adopted the existing CIFAR10 setup that reproduce [70][14]. We changed the model and hyper-parameters to reflect Appendix C.5, and we upgraded the setup to use the latest Flower release at the time (1.7.0).

## D.3 TensorFlow Federated

We used version 0.48.0 in our CIFAR10 and FLAIR setup with TFF because more recent versions removed the Python execution context, and we did not manage to run on GPU with the CPP execution context[15][16]. We adopt the setup from [84] as our starting point, using the commit `a5af8c4433c9ee2d2b1565f1bcc68c89e2000a6b` from 21st February 2024[17]. We changed the model and hyper-parameters to reflect Appendix C.5.

## D.4 FedScale

We used the commit `7ec441c2afa99510535adebe155d89fa8bb2c637` from 18th December 2023 in our CIFAR10 FedScale setup, which was the latest commit as of the release of `pfl-research`. We used the existing CIFAR10 dataset integration available in FedScale and changed the run config to reflect Appendix C.5. We checked that FedScale's CIFAR10 integration was approximately the same as the CIFAR10 setup in `pfl-research`.

Additionally, we did the following modifications in FedScale to more accurately reflect Appendix C.5:

- Disable model saving, which was hard-coded in framework.
- Implement partial participation feature, FedScale samples all users each round!
- Modify the source code in `fedscale/cloud/fllibs.py` to add our model from Appendix C.5.

### D.4.1 Flute

We used the commit `8bfe0854ab293c6226df66856b3d96b39dbe61fe` from 23rd August 2023 in our CIFAR10 FLUTE setup. We adopted the existing FEMNIST setup that FLUTE used to benchmark against FedML[18], and then modified it to use FLUTE's CIFAR10 integration. We changed the model and hyper-parameters to reflect Appendix C.5. FLUTE's configuration files were extensive enough such that we did not need to modify anything in the framework itself. Unfortunately, it was not possible to run multiple training processes on a single GPU, so the only results we have are with $p = 1$ in Table 1.

### D.4.2 System metrics from benchmarks

Figure 7 and Figure 8 show CPU memory, CPU utilization, GPU memory and GPU utilization during training of the CIFAR10 benchmark of each framework. To isolate how performant the frameworks are irrespective of distributed simulation methods, Figure 7 shows experiments utilizing only 1 process for training. **pfl-research** has the highest GPU utilization while simultaneously also having the least RAM usage and one of the lowest CPU utilizations. This is evidence for how our methods 1, 2 and 4 described in Section 4.1 result in a more efficient usage of the hardware. Even though the initialization period before GPU utilization ramps up is larger for **pfl-research**, it is smaller in absolute time since **pfl-research** is much faster. However, FedML has a significant initialization period, 20 minutes, due to a slow data partitioning procedure done before the training starts.

---

[14]`https://github.com/adap/flower/tree/main/baselines/flwr_baselines/flwr_baselines/publications/adaptive_federated_optimization`

[15]`https://github.com/tensorflow/federated/issues/4600`

[16]`https://github.com/tensorflow/federated/issues/4042`

[17]`https://github.com/google-research/federated/tree/master/fedopt_guide`

[18]`https://github.com/microsoft/msrflute/tree/main/experiments/cv_cnn_femnist`

System metrics for CIFAR10 comparison benchmark (1 GPU, 1 process)

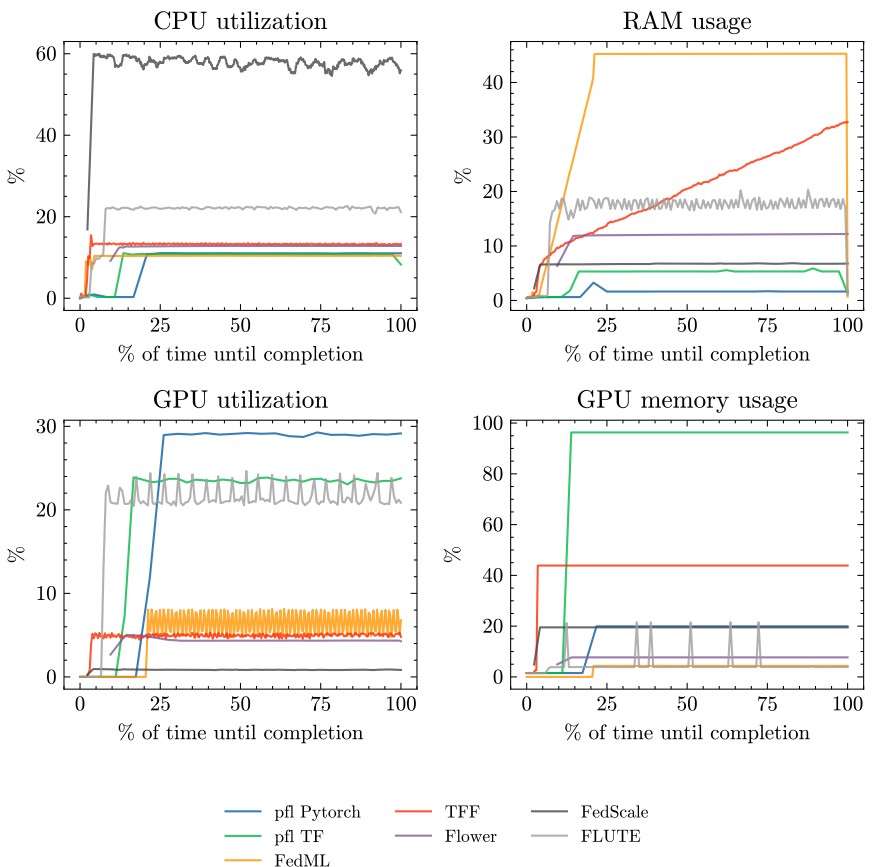

Figure 7: Measurements of CPU utilization, CPU memory usage, GPU utilization and GPU memory usage when running the CIFAR10 benchmark in Table 1 with 1 worker process. The x-axis is the percentage of time until training completes, a relative time measurement.

Figure 8 shows how our methods for efficiently using the GPU also result in a higher multiple of increased GPU utilization from stacking multiple processes training on 1 GPU than other frameworks. Even though the GPU memory usage allows for further increasing the number of processes sharing a GPU for the alternative frameworks, it did not further speed up simulations.

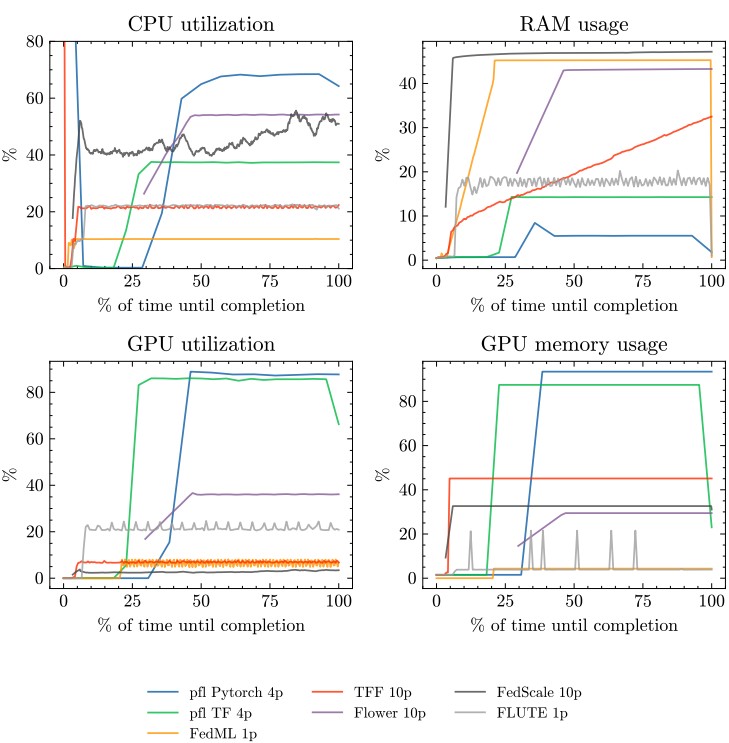

Figure 8: Same type of measurements as in Figure 8 but with the optimal number of processes sharing the single GPU, as also presented in Table 1.

