# OpenReview forum: "$\texttt{pfl-research}$: simulation framework for accelerating research in Private Federated Learning"
_NeurIPS.cc/2024/Datasets_and_Benchmarks_Track — NeurIPS 2024 Track Datasets and Benchmarks Poster_

### Official Review · Reviewer_6DPY · 2024-06-25
**Review for a software suite for simulating federated learning**

**Rating:** 5
**Confidence:** 4

**Review:**

# Strengths

The paper motivates well the need for performant software for simulating federated learning and introduces a new software suite. The software suite brings together existing components (e.g. Horovod for distributed training, TensorFlow/Pytorch for ML) to reduce load on researchers from having to build out this experiment infrastructure themselves which is a valuable contribution to the research community. The paper describes a set of components that can be extension points for others wishing to leverage the existing implementations in `pfl-research` easily and gain access to a potentially large matrix of previously published datasets and algorithms.

The paper provides nice analysis of the strong scaling properties at fixed work loads and increasing compute resources (effectively increasing GPU utilization by adding additional parallel processes), demonstrating ability to "pack" FL clients into a GPU for improved performance. The paper also introduces a mechanism and extension point for scheduling clients in a round across the available users to avoid stragglers, which is critical for improving FL round speed, though the discussion seems largely relegated to Appendix B.4.

# Concerns
The paper does not appear to introduce new _ML performance_ benchmarks or datasets, but focuses on _benchmarks of the software itself_ on existing datasets and models. My initial thought is this maybe a falls bit outside the CfP for the NeurIPS Datasets & Benchmarks track and there may be better venues that focus on performant and scalable ML software. That said, I have still reviewed this paper under the assumption that it is within the theme of the track.

Coming up with methodology for speed comparisons can be very difficult to achieve, especially ensuring strong baselines that provide convincing evidence of SOTA results. I empathize with the authors about the current state of ML software, dependency chains, specialized environments and hardware requirements used across many frameworks. This makes it very difficult to make strong claims about software and its architecture, and not other variables, leading to the speed improvements.

- Discrepancies with previously reported results: FedScale seems to have reported a 1.3 - 3x speed up (depending on cohort size) compared to Flower on CIFAR-10 experiments in FIgure 10 of [1], whereas this paper seems to indicate the opposite (a 3-4x slowdown). Similarly, the FLUTE framework reports a 53x speedup over Flower when using 4 GPUs on a smaller scale experiment [2], a setup which this paper nicely calls out emphasizes the overhead of frameworks, but this paper also seems to have different observations: these two frameworks largely have the same speed. Did the authors see similar discrepancies and know why they seem to arise?

- Weaker baselines: Appendix D.3 explains that the TensorFlow Federated required using an old version, as newer versions were not successful running on GPU. I again empathize with the authors about the difficulty of TensorFlow ecosystem, especially with GPUs, but it seems like a weak baseline to use a 1+ year old version (with hundreds of commits missing). In the past, users (myself included) have found TFF not using multiple-GPUs by default (https://github.com/google-parfait/tensorflow-federated/issues/1029) and requiring additional configuration. Appendix D.3 does not mention whether this was necessary, was this setting also used?

- For all frameworks, was the GPU utilization measured during the experiment, or similar experiments to those performed in Figure 2 for each framework, so that a lack of hardware utilization could be ruled out as a reason for speed differences? Ease/difficulty of achieving high hardware utilization could be something worth mentioning as a comparison point across frameworks; Figure 2 would indicate `pfl-research` achieves high utilization easily by increasing the training processes sharing a GPU.

- If I understand correctly, the code for running the `pfl-research` benchmarks is available under https://github.com/apple/pfl-research/tree/develop/benchmarks, is the code for reproducing the comparisons to other frameworks present in this paper also published? I could not find it at the github repository.

# Opportunities improvement

This paper could become strengthened with a focus on introspective analysis of the different typical scaling measurements, especially along data parallelism and model parallelism. That said, this may be a significant refocusing that could warrant a second review.

## Dimensions of HPC system scaling
When reporting on the speed of HPC cluster performance, it is common to examine the _strong scaling_ and _weak scaling_ characteristics of the system [3]. The paper includes a experiment of a strong scaling nature, where the workload (number of clients per round and number of FL rounds) is held constant, but the available compute capacity (number of parallel processes using a GPU) is increased in Figure 2. In memory bound situations (which appears to be the case here, as the paper mentions a limit to the number of processes can concurrently use a single GPU), experiments for _weak scaling_ (increasing both workload and resources; constant workload per processor) can demonstrate how well the system scales to multiple nodes, and in FL's case, much larger client simulations. This would also be interesting to see how well the communication across machines scales when the simulations become particularly large and the per-round reductions across clients implementation has large impact.

## Deeper analysis of the system
Unlike the sixth reason for improved simulation speed (worker scheduling), which has nice analysis and evidence to support in Appendix B.4, I found the first five reasons (no serialization, one model per GPU, parameters updated in place, not simulating FL topology, and tensors remain on GPU) to lack sufficient and convincing supporting evidence. Profiles of where wallclock time spent during the FL round, similar in vein to those shown in Flower framework's Figure 6 of [4] or Figure 15 of from FedScale [1],  could demonstrate if time is spent in communication across GPUs, ML training, model serialization, etc and would make these claims much stronger.

Relatedly, in Section 3.1, the description of the *Dataset* component points out that in FL many times a client dataset is small enough to simply represent fully in memory. This is a great observation, which was also shown in [5] to have a significant simulation speed impact. Unfortunately the experimental section does not appear to include any analysis on I/O or data preparation. This could be an important hypothesis to rule out? It would be enlightening to see a comparison of frameworks to understand which handle being compute constrained vs I/O constrained.

In general a stronger focus on these dimensions could provide interesting novel insight and side-step the difficulty in setting up rigorous and stronger baseline comparisons to other frameworks.

## Limitations

Mostly for clarification, these may not be limitations.

- Does the software support authoring algorithms such that the clients in the round are computing different local computations (e.g. different model architectures) towards a shared global objective? Reading the code on GitHub, it seems such capability might be possible to simulate with a careful implementation of `FederatedAlgorithm.simulate_one_user` such that different model architectures, or completely different shaped updates could be computed per-client as long as an implementation of an `Aggregator` knew how to deal with them? I'm less familiar with how well this would actually work with the chosen distributed processing mechanism and horovod.

- Does the software support model-parallelism? The paper describes multiple processes sharing a GPU, but does the software allow a single process/client to use multiple GPUs? This could be very important for the recent development of on-device LLMs that frequently and in the 1B+ parameter sizes, such as the experiments reported in [6] from the 2023 Datasets & Benchmarks.

## Points for clarification:

- A `Backend` component is introduced in Section 3 but its not really motivated as to why such a concept or extension point is useful to expose. Only a single implementation, a `SimulatedBackend` exist, but its unclear what other backends might be useful for a software suite that is optimized for simulation and research?
- The experiment section 4.1 provides an explanation for the speed improvement:
  - "no serialization of models": I'm not sure how to reconcile this with section 3.2 description about how communication is necessary in multi-worker scenarios. Is there a method for communicating the models without serialization, or perhaps the statement is referring only to small scale experiments (and not multi worker)?
  - "only one model is initialized and preserved on the GPU at all times" is hard to reconcile with the later statement "FLAIR we can only fit a maximum of p = 3 on the GPU". I would expect when $p>1$ and multiple processes are training client models in parallel on a GPU, it would necessarily require having more than one copy of the model weights (under that understanding, the statement about FLAIR limits on maximum $p$ make a lot of sense to me).

# References

[1] F. Lai, et al. "FedScale: Benchmarking Model and System Performance of Federated Learning at Scale" 2022 39th International Conference on Machine Learning

[2] https://github.com/microsoft/msrflute?tab=readme-ov-file#flower-comparison

[3] https://cvw.cac.cornell.edu/parallel/efficiency/scaling

[4] D. Beutel, et al. "Flower: A Friendly Federated Learning Framework", 2020 https://arxiv.org/abs/2007.14390

[5] Z. Charles, et al. "Towards Federated Foundation Models: Scalable Dataset Pipelines for Group-Structured Learning", 2023 Thirty-seventh Conference on Neural Information Processing Systems Datasets and Benchmarks Track.

[6] M. Abdin, et al. "Phi-3 Technical Report: A Highly Capable Language Model Locally on Your Phone" 2024

**Strengths:**

From the review above:
- The paper introduces a software suite that can reduce load on researchers building up experiment information and speed up research, a valuable contribution to the community.
- The paper provides nice analysis of the strong scaling properties at fixed work loads and increasing compute resources and introduces a mechanism for scheduling clients in a round across the available users motivated by the need to avoid stragglers in FL.

**Additional Feedback:**

No additional feedback above what is above.

**Clarity:**

Summary from the review above:
Generally the paper is well written, a few statements could benefit from additional clarification. More evidence to support claims could strengthen the paper.

**Correctness:**

Summary from the review above:
Concerns about methodology for comparing speed of software suites and discrepancies with previously reported results.

**Documentation:**

From the summary:
- The paper provides appendices of hyperparmaters for all experimental setups, and provides code to reproduce the `pfl-research` results.
- The code provided with the paper at https://github.com/apple/pfl-research does not appear to provide code or further details on how reproduce the speed comparisons to other frameworks, only for reproducing the `pfl-research` results.

**Ethics:**

I do not suspect there are any ethical concerns with the submission.

**Limitations:**

Summary from the review above:
The authors adequately call out future extensions to the library. Some language about ability to support MPMD and model parallelism now or in the future could be helpful.

**Opportunities For Improvement:**

Summary from the review above:
- focus on HPC metrics for reporting on system scaling (weak on strong) properties of pfl-research
- supporting evidence for the rationale given for speed increases (and stronger baselines)

**Relation To Prior Work:**

Summary from review above:
- The paper adequately calls out novel aspects the software suite, including avoids simulating the full FL topology, and worker scheduling.

**Summary And Contributions:**

This paper introduces new software called `pfl-research` for simulating federated learning training in TensorFlow and PyTorch. The paper covers the design and software architecture components which provide abstractions for authoring and an execution runtime. Authoring APIs include components which can be mixed and matched to test different learning algorithms with other techniques (e.g. differential privacy) and provides a few implementations for some commonly used algorithms. The runtime can distribute workloads across hosts as well as GPUs within a host machine, and supports a notion of scheduling client computations on workers within each round of federated learning, with analysis of its effect on runtime. The paper provides experimental results comparing the speed of `pfl-research` against other federated learning software suites using the FLAIR benchmark, and reports the performance of a few different learning algorithms on previously published datasets.

---

> ### Author Rebuttal · Authors · 2024-08-16
>
> > My initial thought is this maybe a falls bit outside the CfP for the NeurIPS Datasets & Benchmarks track
>
> We thank the reviewer for the thorough review under the impression that our work might falls outside of the scope. According to the scope provided in the CFP (https://neurips.cc/Conferences/2024/CallForDatasetsBenchmarks), “this track welcomes ... open-source libraries and tools that enable or accelerate ML research, ..., benchmarks on new or existing datasets, as well as benchmarking tools, ...”. We introduced a new open-sourced library to accelerate PFL research and as a tool for benchmarking new FL algorithms and datasets. We conducted software benchmarks to demonstrate how much acceleration we achieved. We also included PFL performance benchmarks on existing algorithms and datasets. Thus we believe that our work is within the scope of this track.
>
> > Discrepancies with previously reported results: FedScale seems to  ... ... Did the authors see similar discrepancies and know why they seem to arise?
>
> The comparisons reported in [1] and [2] are based on outdated versions of the relevant packages. [1] used Flower 0.17.0 (released on 9/24/2021) and [2] used Flower 1.0.0 (commit on Sep 7, 2022) , whereas we use Flower 1.7.0 (commit on 2/29/2024). Flower has continuously improved its speed along its new releases, while there has not been many improvements to  FLUTE and FedScale during that same time when inspecting the commit activity. Finally, it is of note that the bottlenecks of the various compared FL frameworks are not the same and thus the relative performance of the compared frameworks depends on the hyperparameters, dataset and model.
>
>
> > Weaker baselines: Appendix D.3 explains that the TensorFlow Federated required using an old version...
>
> We thank the reviewer for acknowledging that just setting up TFF for GPU is not trivial, and we agree that this is very unfortunate that we’ve not been able to make more recent versions of TFF work with GPU. According to issues we’re not alone in this. TFF have now removed the issues tab from the repo, which makes this even harder to debug. We cannot even see our own issue on TFF’s Github page. We spent more than twice as much time trying to make recent versions of TFF work than building the setups for the other frameworks. We think it is justified that there is some time limit in building each setup for comparison.
>
> > For all frameworks, was the GPU utilization measured during the experiment ... so that a lack of hardware utilization could be ruled out as a reason for speed differences?
>
> We thank the reviewer for bringing up analysis in system metrics. We have new results regarding this in the general response . This reveals that other frameworks both have a hard time getting their GPU utilization up, and also don’t get that much of a boost in GPU utilization when multiple processes sharing a GPU are added. Keep in mind that the CIFAR 10 benchmarks that always used 1 process/GPU did so because we were not able to make multiple processes training on a GPU work for that particular framework. We will emphasize this in the paper.
>
>
> > is the code for reproducing the comparisons to other frameworks present in this paper also published?
>
> We did not have code for framework comparison released by the submission of this paper because of time and legal process constraints. We will add a link to the code here in the following days for you to view. we will have cleaned it up, made sure it can easily be run, added documentation on how to install and run by the camera-ready deadline date.
>
>
> > ... experiments for weak scaling (increasing both workload and resources; constant workload per processor) can demonstrate how well the system scales to multiple nodes, and in FL's case, much larger client simulations ...
>
> Thank you, this is an excellent suggestion. There was a similar idea displayed in right panel of Figure 3, increasing workload to 50k users/iteration, but your suggestion will be more informative. We will run these experiments (scale number of GPUs on x-axis, while also keeping a constant number of users per GPU trained in 1 central iteration) and add a plot or replace the existing right panel of Figure 3.
>
>
> > I found the first five reasons (no serialization, one model per GPU, parameters updated in place, not simulating FL topology, and tensors remain on GPU) to lack sufficient and convincing supporting evidence.  ...  does not appear to include any analysis on I/O or data preparation.
>
> We agree with the reviewer that a deeper analysis using CPU and GPU profiling of the different frameworks could be insightful. But unlike pfl-research which is very easy to profile due to the replicated processes in a flat hierarchy, the other frameworks were much harder to profile and left out due to time constraints. However, we will commit to adding some results about pfl-research profiling to the appendix. See the post further below for new results on system metrics and a rough description of pfl-research profiling. We believe these additions will give much more insight into how efficient the different frameworks are. “no serialization, one model per GPU ... ” etc is hard to show the impact of in isolation. We argue that these reasons are the majority of contributions toward a much higher GPU utilization as shown in the newly provided figures.
>
>
> > does the software allow a single process/client to use multiple GPUs?
>
> For extremely large models, such as the one in [6], one would typically want to use LoRA or a similar approach to reduce the number of trainable parameters. After all, live deployments of PFL are unlikely to have access to multiple GPUs per user with the exception of large-scale cross-silo PFL. Our focus was primarily on cross-device PFL but we will update the future work section to point out that model parallelism may be desired in some scenarios and in such cases it makes sense to implement model parallelism in pfl-research.

---

> > ### Comment · Reviewer_6DPY · 2024-08-31
> > **Updating my review based on author's reponses**
> >
> > Thank you authors for the discussion and additional evidence gathering, especially the additional GPU utilization plots which provide really strong evidence in understanding why pfl is faster.
> >
> > AC: I would like to increase my review score to 6 based on authors' follow-up, but it appears revisions to my review are locked despite https://neurips.cc/Conferences/2024/Dates saying the discussion period has not ended.
> >
> > In the final version, I would continue to encourage the authors to:
> >
> > - Avoid statements that try to explain the utilization (e.g. model serialization, parameters update in place, topology) without more supporting evidence, or frame the statements as hypothesis for possible follow-ups. Focusing on the GPU utilization being significantly better compared to other frameworks is very well motivated with the new results.
> > - Extend the Limitations section with discussion of model-parallelism (or possibly make a stronger statement upfront how pfl targets cross-device FL, and does not aim to support cross-silo FL directly), and if the current design constrains the implementable algorithms (e.g. heterogeneous models across clients trained in parallel, it wasn't clear from the author's response if this was the case). While all software can always be modified and these all could be extensions to the framework in the future, the paper is covering the state at the time of publication (e.g. the speed comparisons).
> > - Focus on "how/why" the unique/novel insights provided by this collection of tasks (datasets and models) benefits the research community in understanding the problem space of learning algorithms and model architectures. The API design (section 3) and discussion of distributed ML software frameworks are interesting, but there may be enough here to dive more deeply on that independently at a more targeted venue.

---

> > > ### Author Response · Authors · 2024-09-01
> > >
> > > Thank you for revising your score and the additional feedback.
> > >
> > >
> > > > does not appear to provide code or further details on how reproduce the speed comparisons to other frameworks
> > >
> > >
> > > We have now published the FLAIR code for Flower. The code for the FLAIR benchmark comparisons are available at:
> > > pfl-research: https://github.com/apple/pfl-research/tree/develop/benchmarks/flair
> > > TFF: https://github.com/apple/ml-flair/tree/main/benchmark
> > > Flower: https://github.com/grananqvist/flower/tree/flair/baselines/flair
> > > The goal is that a researcher should be able to use the FLAIR dataset in any of the most popular frameworks with the pre-configured baseline setup, and use the same baseline numbers. Then, e.g. algorithm idea A in TFF should be comparable to algorithm idea B in pfl-research or Flower if all built using the initially provided FLAIR setups. To our knowledge, this has not been done before in the area of FL for a realistic dataset.
> > > We will publish the code for running CIFAR10 with each framework soon, and reference to all setups in a dir `pfl-research/publications/system-paper`.

---

> ### Comment · Area_Chair_H6FA · 2024-08-30
>
> Thanks very much for your review. As the discussion is coming to a close, please check the authors' responses and provide your final comments, particularly any specific concerns you may have. Thank you again!

---

### Official Review · Reviewer_daJH · 2024-07-23
**Legitimately useful and efficient software, but the actual data-centric benchmarks could use improvement**

**Rating:** 7
**Confidence:** 4
**Correctness:** See above - though this all seems fine.
**Clarity:** Yes, with some exceptions (see above).

**Review:**

I discuss this in detail primarily in the 2 following sections. In order to avoid repeating myself, I defer the reader to those sections.

**Strengths:**

The software framework itself is clearly beneficial to the federated learning research community writ large. The speedups are significant (self-evidently) but I really did appreciate the scaling behavior discussed in Section 3.2 - details such as the number of clients to simulate within a GPU matter immensely for wall-clock time, and it's nice to see that discussed here.

More generally, the design decisions behind `pfl-research` seem sound. Note that here I am actually not referring to Section 3.1 (which I would argue is more an API description than anything else). Rather, I am talking about the design decisions briefly discussed right before Section 4.2, and mentioned primarily only to give potential explanations for improved runtime of the framework. In fact, I liked this list enough that I wish it were expanded upon (more on that below)

Last, I will note that purely from a dataset perspective, I'm glad that the data-centric simulations involved more interesting datasets than just artificially partitioned versions of something like CIFAR/EMNIST, which are (unfortunately) what most federated learning investigations seem to use. Rather, the Stack Overflow and FLAIR datasets used are legitimately interesting federated datasets, with both scale (number of clients) and heterogeneity that extends beyond just a latent Dirichlet model (or something to that nature). On that note, I was especially pleased with the use of LLM-focused datasets (SA, Aya, and OA). These are likely very interesting from a federated learning + LLMs research perspective, and certainly much more valuable to the research community than yet another benchmark on a small-scale dataset like Shakespeare.

**Additional Feedback:**

I wish to emphasize that I'm a bit torn on this paper. I initially wanted to give it a lower score, due to its focus on performance benchmarking over data-centric benchmarking, but after more reflection and scouring through past archives of Datasets & Benchmarks, I've come to believe that there are some works accepted here that are framework-oriented papers that showcase performance benchmarks, but that combine this with useful, even if succinct, data-centric analyses. I would particularly refer the authors to [Koyamada et al., Pgx: Hardware-Accelerated Parallel Game Simulators for Reinforcement Learning] for a wonderful example of a paper that showcases the speed of their framework, but combines it with important benchmarking experiments that reveal algorithmic and dataset insights (see "Results" in Section 5).

I've since come around to the notion that this paper should likely be accepted - but that it might require some slight re-working of the paper to better discuss things like datasets (and their associated federated statistics) as well as design decisions over API components. Hence I have opted for a borderline accept - with the expectation that the author feedback will push me to a higher score.

EDIT: After the author feedback and reading other reviews, I have raised my score to a 7.

**Documentation:**

The github documentation generally seems really good. The exceptions to this that I am concerned with are all about dataset-specific documentation (e.g. how did the authors partition a dataset across clients) are technically in the code itself, but could be discussed better in the main body of the text (see above).

**Opportunities For Improvement:**

I will preface this with the fact that I think that `pfl-research` is an honestly useful framework, one that I hope can accelerate FL research. That being said, I believe that much of the **data-centric** components of this work - which are the parts most relevant to the intended scope of the NeurIPS Datasets & Benchmarks track (see https://neurips.cc/Conferences/2024/CallForDatasetsBenchmarks) - could be changed in ways that significantly improve the paper. I will break up my suggestions by sub-topic (most of which are data-centric, but not all).

### Missing details in some benchmarks

One issue I want to flag is that for some of the most interesting datasets in the work - Aya, SA, and OA - the actual paper provides almost no details on the dataset, and how to use it with `pfl-research`. Notably, these three are "centralized" datasets (primarily used for non-FL tasks) that have been partitioned (both in heterogeneous and homogeneous ways) across clients. The paper simply states that "Aya and OpenAssistant datasets have inherent user partition" (L836-837). The paper is likely referencing, for example, the `user_id` field in the huggingface dataset (e.g. https://huggingface.co/datasets/CohereForAI/aya_dataset) but (1) this is not clear and (2) more importantly, the user is given no greater insight into the dataset. How many users are there? How do they vary in number of examples or sequence length of examples? This is especially important in the case of SA, which has no "canonical" user partitioning. The appendix states that they partition it in an IID fashion, but do not specify across how many users. Poking through the github, I discovered that the code suggests that  they sample the number of examples via the aforementioned Poisson **without replacement** and proceed **until the dataset runs out of examples** which governs the number of clients. These are all details that I wish were in the actual body of the paper - along with things like dataset statistics, as they are so data-centric.

### Uncertain takeaway from the data-centric benchmarks

The next way the paper could improve is by giving some form of data-centric benchmarks that actually provide some insight into things like algorithmic choices, or how things like dataset statistics influence performance. The reader takeaway of Tables 3 & 4 in the work is somewhat opaque to me. In fact, due to how things were tuned, the authors state "Therefore, no conclusions should be made about the performance of the algorithms relative to FedAvg." While I think it's great to be honest about this, I find myself wishing that the authors had designed the benchmarks in a way that the reader could take home some kind of algorithmic or dataset-oriented takeaway. Tying in to my issue with lack of clarity on datasets above, one question the authors might begin to pose is how algorithm performance changes as a function of various statistical properties of the dataset, especially given the fact that (other than SCAFFOLD), the algorithms considered all perform very similarly.

### Data-centric versus performance benchmarks

I wish to preface this section with the fact that my interpretation of this track's scope may not be the same as the authors', but I can only review according to my own interpretation. To preface this, recall the data-centric benchmarks discussed above (Section 4.3). The authors' explicit intention seems to have been to show what kinds of research could be done with `pfl-research`. But this is where I want to bring up the scope of the Datasets & Benchmarks track again: the focus really is on data-centric benchmarks, as opposed to performance benchmarks. So while the performance benchmarks are great (really, they're well done), the benchmark that I find most lacking (at least in terms of actionable insights) are the data-centric ones. This is all to say that I wish there was a bit more emphasis on how `pfl-research` can enable new insights into data-centric machine learning (and at a greater velocity). I understand that wall-clock time of experiments is its own benefit, so my opinion here is somewhat weakly held. But it is something that jumped out at me when reading this. This is why I harped on the lack of dataset details above - those details, potentially combined with some kind of interesting ablation, may offer much more insight than benchmarking on multiple similarly-performing algorithms.

### Discuss design decisions instead of API specifics

In short, I found Section 3.1 a bit strange and not helpful. By contrast, I found the discussion right before Section 4.2 to be incredibly useful, and wish it were discussed more instead.

In more detail, Section 3.1 reads more like an API doc than anything else. While I understand that it gets at the flexibility of the API (e.g. the various ways in which a user can alter behavior) I honestly think it goes too much into the weeds. For example, the protocol-like definition of an algorithm in L134-139 does not really inform the reviewer why `pfl-research` is useful. Moreover, some of Section 3.1 is presented in a way that only says what the component does, not why it was designed in the way it was (e.g. why are post-processors implemented in reverse order? why is it important that the reader understands that so early on in the paper?)

By contrast, I thought the design decisions right before Section 4.2 were far more valuable in insight, and could use more exposition. How much benefit is incurred by avoiding model serialization, for example? What does it mean to use tensors "end-to-end"  (L246)? How is load balancing done? Personally, I would've liked to see this be the focus of the main body of the paper - higher level ideas that differentiate `pfl-research` from other frameworks, and why they matter for performance.

**Relation To Prior Work:**

While this section is generally fine, the authors tend to make some very broad statements that they could state more accurately by narrowing down. For example, they state that "Most existing FL frameworks do not provide a simple way to plug [sic] DP mechanisms into existing FL algorithms while ensuring that the parameters for DP are consistent with the experimental setup." What do you mean by "ensure the parameters...are consistent"? Do you mean that things like the actual noise multiplier for a given differential privacy experiment, when run for the given number of rounds, will incur the desired privacy level? Do you mean that the framework should error our (or something similar) if incompatible noise multipliers are given? Or do you mean that the library should do the noise accounting and simulation simultaneously? I think that by scoping this down, you can make a stronger statement here.

Second, while I support the authors discussion of the limitations of commonly-used datasets in the federated learning literature (L84-93), the metric used here is suspect: The authors point to a specific workshop on FL. It is not clear to me how statistically significant this is in the grand scheme of FL research (for example, maybe that year the workshop focused more on theory than on empirical analysis?) I completely agree with the notion that the authors discuss here - I just wish to emphasize that there may be a better way to quantify this.

I also wish that the authors had highlighted efforts to create, curate, and disseminate more realistic federated learning datasets, including (roughly in order of year):

* Caldas et al., "LEAF: A benchmark for federated settings."
* Hsu et al., "Federated visual classification with real-world data distribution."
* He et al., "FedML: A Research Library and Benchmark for Federated Machine Learning."
* Lin et al., "FedNLP: Benchmarking Federated Learning Methods for Natural Language Processing Tasks."
* Lai et al., "FedScale: Benchmarking Model and System Performance of Federated Learning at Scale."
* Charles et al., "Towards Federated Foundation Models: Scalable Dataset Pipelines for Group-Structured Learning."

Note that while some of these are cited (especially as many are benchmarking frameworks), their specific efforts to promote and make accessible large-scale and realistic datasets is not really mentioned in this work.

**Summary And Contributions:**

This paper introduces a software framework (`pfl-research`) designed to make it easy and efficient to conduct research on federated learning. It discusses the various improvements it provides over other federated learning research simulation frameworks. The paper discusses the API & system design of the framework, and then jumps into simulation results.

Essentially, the paper gives three sets of benchmarking results. The first set benchmarks wall-clock time needed to attain a specific accuracy threshold of `pfl-research`, compared to other federated learning simulation frameworks. The second set benchmarks the scaling behavior of `pfl-research`, especially in distributed training settings. The last set of benchmarks compares a variety of federated learning algorithms on a variety of datasets. The first 2 sets of benchmarks generally show that the framework is faster than others, and exhibits good scaling properties. The last set of benchmarks is conducted mainly to show that it is possible to do such an evaluation (more on that below).

---

> ### Author Rebuttal · Authors · 2024-08-16
>
> > ... Aya, SA, and OA - the actual paper provides almost no details on the dataset, and how to use it with pfl-research. ...
>
>
> We thank the reviewer for pointing out the missing details in the dataset description. We also provide the details of the dataset description and statistics in below. In the updated version of the paper, we will add those details in the main body.
>
> Stanford Alpaca (SA) contains 52,002 instruction-following data. As SA has no intrinsic user partition, we partition the data into users in an I.I.D fashion.  We first sample the length $L$ of each user dataset using Poisson distribution with expectation of 16 data per user, and then assign the unused $L$ data points to the user. The partition stops until all data points are assigned.
> Aya contains a total of 204,112 human-annotated prompt-completion pairs, where each label has the identifier of the annotator. We partition the dataset into users by the provided annotator identifier such that the data is non-I.I.D. We constraint the maximum data size per each user to 64, and if an annotator has more than 64 pairs, we evenly split this annotator’s data into smaller subsets of size 64.
> OpenAssistant (OA) contains more than 120,000 conversational messages between human and AI assistant pairs. We collect 85,318 pairs of user inputs and assistant response with associated user identifier. We partition the dataset into users in a non-I.I.D fashion using the provided user identifier.
>
> | Dataset | # of users | # of val users | # of samples per user (mean +/- std) |
> |---------|------------|----------------|-----------------------|
> | Aya     | 4303       | 216            | 47 +/- 25             |
> | SA      | 3267       | 327           |      16 +/- 4              |
> | OA      | 14300      | 1430          |  6 +/- 32                  |
>
>
> >  I find myself wishing that the authors had designed the benchmarks in a way that the reader could take home some kind of algorithmic or dataset-orixented takeaway.
>
> We are glad that the reviewer feel this way. The benchmarks are well tuned, but common hyperparameters were only tuned for FedAvg, and re-used for the other algorithms while tuning the algorithm-specific hyper-params. We removed some conclusions because of this to be extra conservative. We can simply include them but still mention the caveat.
>
>
> * Banded matrix factorization mechanism, also known as DP-FTRL when applied to FL, outperforms Gaussian mechanism with PLD moments accountant on StackOverflow, with a big 10% relative improvement.
> * SCAFFOLD, a popular algorithm according to citations, does not perform better than FedAvg on any of our benchmarks. [63] also report that SCAFFOLD consistently perform worse than FedAvg.
> * FedProx is known for performing well on heterogeneous data partitions, but this is only marginally reflected in the FLAIR benchmark results. It is worse than FedAvg baseline on FLAIR IID, but better on FLAIR with natural heterogeneous client partitions.
>
>
> > Data-centric versus performance benchmarks
>
> Please see the our post on "on the relation of pfl-research to data-centric benchmarking" in the general response.
>
>
> > In short, I found Section 3.1 a bit strange and not helpful. By contrast, I found the discussion right before Section 4.2 to be incredibly useful, ...
>
> We thank the reviewer for the suggestions for the organization in Section 3.1. In the updated version of the paper, we will reduce the size of this section and include more discussion on why the design is chosen and how it would impact the overall performance of pfl-research.
> We acknowledge the reviewer’s feedback that the API and class structure is too detailed. This stems from our study of pros and cons of FL frameworks and feedback from researchers internally and externally (some most recent from ICML ’24 expo talk).
>
>
> >  "Most existing FL frameworks do not provide a simple way to plug [sic] DP mechanisms into existing FL algorithms while ensuring that the parameters for DP are consistent with the experimental setup." What do you mean ...
>
> Our Speed at which DP can be applied: don’t need to cast to numpy/use CPU. All DP mechanisms run on GPU.
> Integration is superior: Breadth of privacy mechanisms and accountants integrated in framework. Flower does not well support DP. TFF does provide DP experimentation for PFL, through use of TF privacy.
>
>
> > The authors point to a specific workshop on FL. It is not clear to me how statistically significant this is in the grand scheme of FL research
>
> Thank you for your feedback, We did a survey about most recent publications at NeurIPS 2023, ICLR 2024, ICML 2024. There are 162 publications that provide simulation results in federated learning. 142 out of 162 publications provide results on 1-100 clients, while only 6 out of the 162 publications consider more than 1000 clients for the empirical analysis. This is in line with our observation in the original submission, and emphasize the our fast simulation framework could accelerate the future pfl research for more realistic scenarios with more clients. We will add such discussion in the revised manuscript.
>
>
> > ... their specific efforts to promote and make accessible large-scale and realistic datasets is not really mentioned in this work.
>
> we strongly agree that the efforts on promoting and creating realistic datasets for large-scale PFL simulations are critical and we see pfl-research as an important contribution that enables efficient PFL simulations with these datasets. We will include the 3 above citations that were missing into the paper and explicitly point out the connection between the efforts to create realistic, large-scale datasets and pfl-research in Section 2.
>
>
> > I wish to emphasize that I'm a bit torn on this paper. ...
>
> We appreciate that you explain your thought process here about the relevance to the track. We have provided a response in our general rebuttal under title "on the relation of pfl-research to data-centric benchmarking".

---

> > ### Comment · Reviewer_daJH · 2024-08-28
> > **Revisiting my original review**
> >
> > Hello all. Thanks for the detailed comments. In short, I will be raising my score to a 7. That being said, I do want to emphasize (1) changes suggested above that I think are crucial to warrant this score and (2) some drawbacks that I don't think the authors have fully addressed, but that don't do enough to make me recommend rejection.
> >
> > **Crucial changes**
> >
> > * Details on the Aya, SA, and OA datasets and their associated statistics.
> > * Clear discussion around the hyperparameter tuning that was done, the limitations, and what conclusions the reader should (and should not) draw from the results.
> > * Reducing the level of detail in Section 3.1 - While the pros and cons of the system are important, API-level details are probably best left to the github level.
> > * Either add evidence supporting them, or remove of the system benefits in Section 4.2. Reviewer 6DPY makes a great point here - these are really specific claims that largely have no justification. If a benefit is difficult to show, then I think that claiming it without justification is problematic.
> >
> > **Some reservations I still have** - Again, just a reminder that these are not blocking, I still recommend acceptance. This is largely feedback for the authors.
> >
> > * I still believe that the design of the benchmarks emphasized breadth of algorithms over actual data-specific insights, and that this is a drawback of the work.
> > * If the goal of the work is to convince a Datasets & Benchmarks reader to use the framework, then some kind of demonstration as to how the framework led to some kind of insight that was not known before would be very useful. As I stated in my review,  [Koyamada et al., Pgx: Hardware-Accelerated Parallel Game Simulators for Reinforcement Learning] do this nicely in section 5 of their work.

---

### Official Review · Reviewer_YXDe · 2024-07-25
**Simulation-focused FL framework emphasizing simulation speed**

**Rating:** 8
**Confidence:** 4
**Correctness:** Most claims seem correct.

**Review:**

The paper is generally clear and original. It adds yet another FL simulation framework that is research simulation focused and could be potentially helpful in FL-related research.

**Strengths:**

- Comprehensive system design and engineering details which is rarely seen in research-focused framework construction, impl is also provided
- It considers privacy and its influence in FL training, which is usually a lacking feature in similar frameworks
- Speed up seems promising and the performance discussion is reasonable

**Additional Feedback:**

N/A

**Clarity:**

The paper is well-written but might be too focused on engineering / system design claims, while concrete technical novelty for such details are not justified.

**Documentation:**

N/A

**Ethics:**

No.

**Limitations:**

- The lack of deployment-related consideration (e.g. device availability simulation / comm env simulation) prevents the framework from being useful in system-related algorithm exploration
- Lack of profiling on the performance influence from privacy machanism (e.g. does adding DP influence simulation time / influence convergence speed, then influence training round number, etc.)
- Lack support/discussion in cross-silo setting

**Opportunities For Improvement:**

- Influence of secure aggregation in terms of simulation speed is not well-discussed, although mentioned in the introduction
- More frameworks might be considered for system performance (e.g. Fate / FedML / Flute / FedTree / etc.) while currently only TFF and Flowers are main targets
- Discussions of multi-host multi-gpu environment might be interesting

**Relation To Prior Work:**

Existing federated learning systems and benchmarks are well-discussed. Discussion about other FL simulation systems and comparison is insufficient.

**Summary And Contributions:**

The paper discuss a new Python FL framework as well as new bundle of datasets for FL simulations. It has comprehensive system engineering design and achieve significant speed up compare with some other FL frameworks. It support various settings as well as various training algorithm with different ML backend. It also provide a comprehensive scalability discussion in different mode of usage.

---

> ### Author Rebuttal · Authors · 2024-08-16
>
> > Influence of secure aggregation in terms of simulation speed is not well-discussed, although mentioned in the introduction
>
>
> We thank the reviewer for this observation. The lack of discussion here, is because simulations have no impact with secure aggregation, as the purpose is summing or averaging as an opset, over the gradients and metrics over central iterations. The communication I/O cost due to newer, efficient protocols like PINE (Rothblum, Talwar et al) is outside of the scope of core FL simulations that this paper introduces.
>
>
> > More frameworks might be considered for system performance (e.g. Fate / FedML / Flute / FedTree / etc.) while currently only TFF and Flowers are main targets
>
>
> Thank you for the comment. While we recognize that there are more frameworks available than what we included in the benchmarking, we want to remind the reviewer that we did not just focus on TFF and Flower. We did the large-scale simulation comparison using FLAIR in Table 2 with only TFF and Flower because they are most widely used for research and performed okay on the CIFAR benchmark. In Table 1 we do compare performance of more frameworks, including some you mentioned: FedML, TFF, Flower, FedScale, FLUTE.
> We’re also cognizant of NVFLare, but they’re now integrating with Flower (https://arxiv.org/abs/2407.00031) and hence our benchmark coverage will indirectly extend to it.
>
>
> > Discussions of multi-host multi-gpu environment might be interesting
>
>
> Thank you for the comment. Indeed, simulations in a multi-host, multi-GPU environment are relevant for large-scale simulations and can be done using Horovod. We will point this out explicitly in the introduction where we discuss Horovod. Results presented in Figure 3 actually use multi-host simulations for the cases where #GPUs>8. We will clarify this in the paper, and you can see the GPU hours start to get slightly less efficient at >8 GPUs. Reviewer 6DPY suggested adding results for weak scaling, which we will add to the paper. This will show some further results on what happens when scaling up beyond 8 GPUs.
>
>
> > The lack of deployment-related consideration (e.g. device availability simulation / comm env simulation) prevents the framework from being useful in system-related algorithm exploration
>
>
> Thank you for the comment. The paper and the underlying pfl-research framework focus on PFL simulations for accelerating and enabling research in FL. PFL simulations are important for deployment of live PFL systems, but the deployment of live PFL systems is outside the scope of this work. The framework focuses more on data-centric evaluations of algorithms.
>
>
> > Lack of profiling on the performance influence from privacy machanism (e.g. does adding DP influence simulation time / influence convergence speed, then influence training round number, etc.)
>
>
> Yes, we agree that it is important to highlight any slowdown in simulation speed that DP might cause. Time to converge on FLAIR with central DP is 1.93h with pfl==0.2.0 (that is only 9% slower than without DP).  We will add an entry “pfl-research 0.2.0 (PyTorch) with Central DP” in Table 2  with this number.  Regarding the performance of FL with and without DP for a fixed number of central iterations, we provide the metrics with and without DP in Table 3 and Table 4 respectively, both with the same number of central iterations. Tables 8-11 also show that we do not modify the number of iterations when enabling DP.
>
>
> > Lack support/discussion in cross-silo setting
>
>
> We agree with the reviewer that we do not put much focus on cross-silo FL. As described in future work, we do provide support for it but we don’t have any example or benchmarks yet. We thought this was less important because cross-silo is a strictly easier setup to build efficient simulations for, which existing tools do an okay job with. We will rely on the external community to contribute more cross-silo benchmarks and features if they deem it important.
>
>
> > The paper is well-written but might be too focused on engineering / system design claims, while concrete technical novelty for such details are not justified.
>
>
> We thank the reviewer for highlighting the strength of engineering and systems design aspect. The technical novelty of this work is primarily in the efficiency/performance of the design that enables testing and benchmarking FL algorithms and large datasets that were previously infeasible. Another important novelty aspect that follows from the design is the ease with which one can implement and test new algorithms (e.g. new optimizers, DP).
>
> > Existing federated learning systems and benchmarks are well-discussed. Discussion about other FL simulation systems and comparison is insufficient.
>
>
> We thank the reviewer for highlighting the thorough discussion of existing FL systems and benchmarks. We are not sure what the reviewer meant by insufficient discussion of other FL simulation systems and comparisons; we included 5 specific FL simulation systems and discussed what we believe are the key reasons that make pfl-research more efficient as was demonstrated by experimental results. Table 1 is, to our knowledge, the largest FL framework speed comparison done so far for a setup with at least 1000 users and a 1.6M parameter model, and even though there are only 3 entries in Table 2, it is still the largest speed comparison of frameworks using a benchmark with >50k users and a 50M parameter model on a realistic FL dataset. E.g. UniFed [1] data and model setups are much smaller than ours.
>
> [1] X. Liu et al. “UniFed: A Benchmark for Federated Learning Frameworks”

---

### Official Review · Reviewer_Y2HC · 2024-07-25
**A federated learning framework with superior scalability, efficiency and differential-privacy tools**

**Rating:** 6
**Confidence:** 3
**Correctness:** yes
**Clarity:** yes

**Review:**

**Pros**

- The superior efficiency in large-scale simulation enables researchers to study FL in a more realistic FL environment.
- The framework supports multiple backends like pytorch and tensorflow
- It's convenient to conduct the impact of DP in this framework.

**Cons**
- The comparison of memory cost of the simulations was missing, where the memory may quickly increase with the number of spawned processes.
- It seems that the parallelism is arranged at the client-level. Is there any command-level (e.g, running different configurations)  parallelism in this framework?

**Strengths:**

See pros in review.

**Additional Feedback:**

Please see Cons in review.

**Documentation:**

yes

**Limitations:**

yes

**Opportunities For Improvement:**

Please address the cons in review

**Relation To Prior Work:**

yes

**Summary And Contributions:**

This work proposes a general FL framework for efficiently simulating FL on large and realistic FL datasets. The contributions are summarized as follows:

- The framework is 7x-72x faster than existing frameworks under cross-device FL settings.
- The scalability of the framework enables researchers to conduct FL in a more realistic setting.
- Popular tools for differential privacy were integrated into this framework to provide convenient privacy analysis.

---

> ### Author Rebuttal · Authors · 2024-08-16
>
> > The comparison of memory cost of the simulations was missing, where the memory may quickly increase with the number of spawned processes.
>
>
> We thank the reviewer for bringing this up. These are important measurements to include in the paper. We have submitted a separate post below regarding CPU & GPU memory and utilization. We made one figure using 1 training process only and 1 figure using the number of processes that were used in Table 1 (FedML and FLUTE still use 1 process here because they do not support multiple processes sharing a GPU, or it was too difficult to get working given an equal amount of time spent setting up the benchmark for each framework.)
>
>
> > It seems that the parallelism is arranged at the client-level. Is there any command-level (e.g, running different configurations) parallelism in this framework?
>
>
> We thank the reviewer for pointing out this issue; it is indeed important to not only parallelize a single simulation run but also to allow running multiple runs in parallel e.g. for hyperparameter tuning. In our documentation (https://apple.github.io/pfl-research/guides/simulation_distributed.html) we show how easy it is to run and scale up distributed simulations using horovod, mpirun or the native tf.distributed and torch.distributed. These can be combined with workload managers (e.g. slurm) that depend on the specific cluster ecosystem, which provide tools for (1) resource allocation for the individual simulations, (2) starting, executing, and monitoring work, and  (3) managing a queue of pending work. We will point this out in the introduction after mentioning Horovod in the introduction section.

---

> ### Comment · Area_Chair_H6FA · 2024-08-30
>
> Thanks very much for your review. As the discussion is coming to a close, please check the authors' responses and provide your final comments, particularly any specific concerns you may have. Thank you again!

---

### Author Rebuttal · Authors · 2024-08-16

We want to thank all the reviewers for all the detailed feedback. There were many good suggestions we will incorporate to increase the quality of the paper.
There were two topics that multiple reviewers mentioned, so we provide a general rebuttal regarding those below:


## on the relation of pfl-research to data-centric benchmarking

First of all, note that according CFP (https://neurips.cc/Conferences/2024/CallForDatasetsBenchmarks), this track “welcomes ... open-source libraries and tools that enable or accelerate ML research, ..., benchmarks on new or existing datasets, as well as benchmarking tools, ...”. Our paper introduced a new open-source library to accelerate PFL research, and to benchmark new FL algorithms and datasets. We conducted a number of experiments to demonstrate how much acceleration was achieved, showing that we can achieve speedups of an order of magnitude or more (see Tables 1 and 2). That means that our work is clearly within the scope of this track.

In addition to this, our paper provided examples that demonstrated the new data-centric insights that become possible thanks to the efficiency of our framework. Table 3 provided an analysis of several FL algorithms (FedAvg, FedProx, AdaFedProx, SCAFFOLD) on 6 datasets with IID and non-IID data. Table 4 extended the analysis to include 2 privacy mechanisms (Gaussian moments accountant and banded matrix factorization). Many of these algorithms have not yet been evaluated on such benchmarks.

Our framework provides a more efficient way of benchmarking FL with DP without the need for cohort sizes in the order of 1000s (see Appendix C.4). Without this, PFL simulations with strong DP guarantees may often become infeasible for large-scale benchmarks. The importance of this approach was demonstrated e.g. in our work on ASR [9].

In the readme of pfl-research, we reference a weights-and-biases (W&B) report that can effectively summarize the benchmarks at https://wandb.ai/pfl/benchmark-algos/reports/PFL-algorithms-benchmark-results-PyTorch---Vmlldzo2NDYwODMx?accessToken=cnq7duqx3bwfgcsjl3v6h8n4c95lfyeqpjtmlmm6gfgxirrxxvnhoxlnxoa2jeft. This reports will serve as data-centric baselines for all future enhancements. As a community, we strive to continually improve model performance in various domains (LMs, vision, recommender systems, FMs, etc). These can be published in W&B reports and contributed to by the broader community. Our PFL framework enables and accelerates research in the PFL field by providing a comprehensive software performance benchmarks.

## on missing system metrics

The reviewers asked about system level metrics on multiple occasions. We provide 2 new figures showing CPU & GPU utilization and memory allocation for the frameworks we compared on cifar10. We will update the paper with these results. See attached PDF.

Conclusions:

* Only pfl-research can make efficient usage of sharing a GPU for multiple processes in the case of small models.
* For single-process, pfl-research has highest gpu utilization while also having among the lowest cpu-utilization. This supports the claims that pfl-research make efficient usage of the GPU: e.g. using tensors end-to-end in the outer training loop of FL, minimizing the number of new models initialized into memory by making all model modifications in-place
* FedML take extra initialization time because it splits up datasets very slowly making it not usable if you have >10k users.
* Looks like there is some memory issue for TFF not related to GPU memory in this setup which was adopted from the google-research repo.

---

### Decision · Program_Chairs · 2024-09-26

**Decision:**

Accept (Poster)

**Comment:**

The proposed pfl-research is a fast, modular, and easy-to-use Python framework for simulating FL. After thorough rebuttal processes, all reviewers have provided positive feedbacks (Reviewer 6DPY missed the rebuttal deadline, but claimed to increase the score from 5 to 6).  It is recommended that the authors continue to maintain and enhance the actual software and benchmark according the reviewers' comments.